# GAMBIT: A GRAPH-STRUCTURED AND DECISION-AWARE BENCHMARK FOR MOBILE GUI TASKS

## ABSTRACT

Mobile GUI agents powered by LMMs can perceive screens and follow instructions, yet existing benchmarks largely target short, linear workflows and step-level accuracy, offering limited insight into long-horizon planning and branching tasks. We present GAMBIT, a graph-structured, decision-aware benchmark comprising 830 task episodes and 11,345 actions across 35 applications on Android and iOS. Tasks are organized into Sequential, Conjunctive, Conditional, and Hierarchical workflows with dual-level annotations, capturing realistic multi-step and branching scenarios. To move beyond step metrics, we introduce weighted longest common subsequence for length-sensitive progress and decision accuracy for branch correctness. Evaluations on 7 diverse agents show that GAMBIT induces a substantial accuracy drop compared to prior datasets, with success rates falling below 5% on 6–8 step tasks and branch accuracy averaging 38%, underscoring weaknesses in conditional reasoning. By systematically exposing these failure modes, GAMBIT provides a challenging, diagnostic testbed for advancing decision-aware mobile GUI agents. Our code and dataset are available at: https://anonymous.4open.science/r/GAMBIT-40BB/.

## 1 INTRODUCTION

Recent advances in Large Multimodal Models (LMMs) have substantially improved capabilities in visual content understanding Yin et al. (2024), following complex instructions Wen et al. (2024a); Qin et al. (2024), and planning multi-step tasks Li et al. (2024a), paving the way for autonomous agents in real-world applications. Within this broader landscape of agents, ranging from tool-use and function calling systems Fan et al. (2024); Shen et al. (2023), to embodied agents and domain-specific assistants, graphical user interface (GUI) agents have emerged as a practical and versatile paradigm. By perceiving screen content and executing structured actions, GUI agents can operate existing software environments Liu et al. (2024b); Zhang et al. (2024a); Hu et al. (2025). Among them, mobile GUI agents Chai et al. (2025); Jiang et al. (2025) have attracted increasing attention due to their wide relevance across everyday applications and the ubiquity of smartphones. Accordingly, growing efforts have been devoted to developing mobile GUI agents that simulate user interactions and automate routine tasks on devices Ye et al. (2025); Tang et al. (2025).

Systematic evaluation of mobile GUI agents is therefore critical, both to assess current capabilities and to identify limitations that guide future development. Existing benchmarks, however, vary widely in their scope and emphasis. Li et al. (2025) focus primarily on GUI element grounding and visual understanding, providing the perceptual foundation for agent actions. Rawles et al. (2023) and Zhang et al. (2024b) extend to single-step and multi-step instruction execution, with the latter incorporating step-wise reasoning. Li et al. (2024b) highlights task difficulty by varying instruction length and step count, and Lu et al. (2024) pushes toward cross-app scenarios that require layout adaptation and app switching. More recently, Zhang et al. (2025c) emphasizes Chinese application layouts and bilingual instructions beyond English-centric focus of prior datasets.

Despite these advances, current datasets and benchmarks still exhibit three major limitations. 1) **Limited instruction diversity.** Most benchmarks expand tasks by template substitution or linear concatenation of short instructions Chen et al. (2024); Lu et al. (2024). Such construction not only produces semantically repetitive tasks, but also reflects the biases of a small set of annotators' brainstorming habits. Moreover, action descriptions often reuse the same phrasing and constraints,

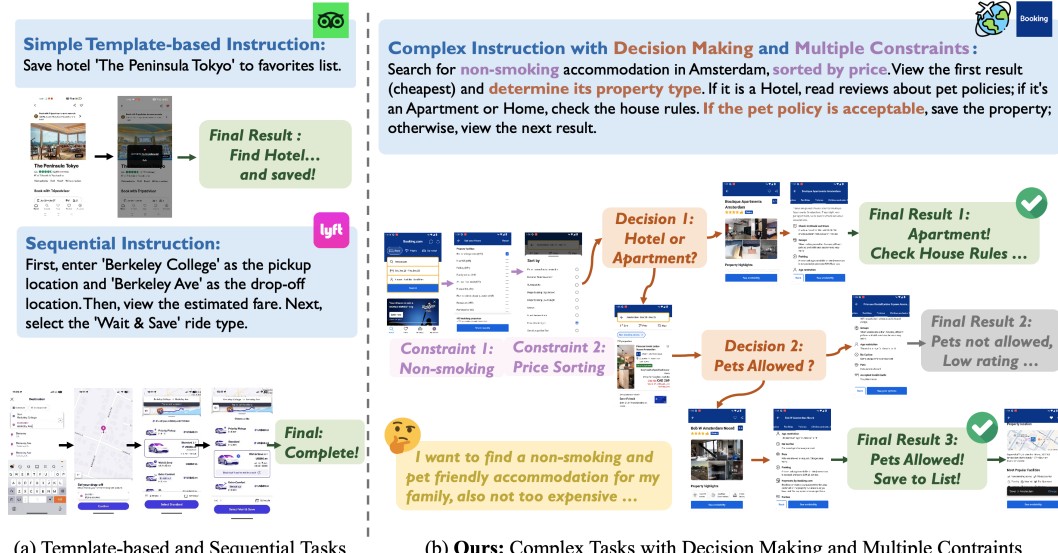

(a) Template-based and Sequential Tasks      (b) **Ours:** Complex Tasks with Decision Making and Multiple Contraints

Figure 1: Illustration of conventional step-based tasks with linear chains versus our proposed graph-structured tasks incorporating decision-making and branching conditions.

failing to capture the richness of real-world user instructions. 2) **Overly sequential workflows.** Existing datasets predominantly formalize tasks as linear step-by-step chains with few constraints, overlooking decision-aware behaviors involving conditional branching, fallback strategies, or multi-condition constraints. In realistic scenarios, users routinely adapt their workflows (e.g., *"purchasing a high-priced item if the cheaper one is out of stock"*, or *"adjusting travel plans based on weather forecasts"*). 3) **Inadequate evaluation protocols.** Current metrics focus on step-level correctness or holistic task success rate. These measures neither weight longer workflows appropriately nor capture decision accuracy at branching nodes, leaving gaps in assessing agents' robustness to task complexity and structural variability. These limitations hinder systematic evaluation of mobile GUI agents' capabilities in long-horizon, decision-aware, and cross-context scenarios.

To address these limitations, we introduce **GAMBIT**, a complex benchmark for evaluating mobile GUI agents on long-horizon and decision-aware tasks. We propose a human-LLM collaborative **atomic instruction and constraint pipeline**, which composes diverse atomic actions into **Graph-structured Complex Tasks** spanning Sequential, Conjunctive, Conditional and Hierarchical work-flows. A mixture-of-generators task construction and dual-layer quality control process ensures both executability and diversity. GAMBIT comprises 830 task episodes with 11,345 actions steps, covering 35 applications across 7 mainstream categories on both Android and iOS platforms, with dual-level annotations and an average depth of 13.3 steps. To complement the dataset, we design decision-sensitive evaluation metrics that weight long-horizon steps and explicitly measure branch-ing accuracy, providing a more faithful reflection of agent capability than conventional step-level exact-match or success-rate metrics. These establish GAMBIT as a challenging and diagnostic benchmark for analyzing current mobile GUI agents, exposing their bottlenecks and guiding future progress in agent design. Our key contributions are as follows:

1. We release **GAMBIT**, the first benchmark targeting long-horizon, decision-aware mobile GUI tasks, covering diverse application scenarios that are representative of everyday usage.

2. **Principled construction pipeline:** we design a collaborative pipeline that expands atomic in-structions with constraints and composes them into graph-structured tasks, ensuring both realism and semantic diversity.

3. **Decision-sensitive Evaluation:** we propose metrics that go beyond step-level exact match and global success rate by weighting long-horizon steps and explicitly measuring branching accuracy.

4. **Comprehensive evaluation:** we conduct systematic experiments across 7 general-purpose, mobile-specialized, and reasoning-oriented agents, revealing critical bottlenecks and providing di-agnostic insights for future model development.

## 2 RELATED WORK

Prior efforts in mobile GUI benchmark construction can be grouped into three following paradigms. 1) Human Handicraft Designed Instructions: AITW Rawles et al. (2023) categorizes tasks based on the number of action steps into multiple-step and single-step categories. Multiple-step tasks are obtained through human annotation, technical documentation, and LLM-enhanced instructions, while single-step tasks are derived by extracting shorter action sequences from longer action sequences. AITZ Zhang et al. (2024b), built upon AITW, further filtered out incorrect or mismatched instructions and utilized GPTs to assist in proofreading task instruction semantic descriptions. Other mainstream datasets and benchmarks Rawles et al. (2024); Sun et al. (2022); Li et al. (2020); Wen et al. (2024b); Lee et al. (2024) widely adopted such human written instructions to simulate real-world user cases by either crowd sourcing or mimicking daily user case and routine. 2) Extracted and generated from large-scale dataset or website Liu et al. (2025a); Chai et al. (2024); Zhang et al. (2025b): Android-arena Xing et al. (2024) identify webpage and descriptions related to application functionality through web retrieval, then store these software descriptions in a vector database, and finally utilize an LLM to reconstruct app tasks instructions from them. 3) Expand based on existing instructions and template replacement: GUI-Odyssey Lu et al. (2024) through manually brainstorming instruction task templates, through template substituting the application name or action description, quickly expand current instruction to a large-scale dataset and finally rewrite with GPT-4. SPA-Bench Chen et al. (2024) first start with single-step instruction and through expanding one action at a time to obtain long chain instruction and finally achieve difficulty rising. Despite these efforts, existing benchmarks exhibit limitations: heavy reliance on human annotation or template substitution, limited instruction diversity, and a bias toward chain-like workflows. Consequently, they fall short of simulating the decision-making and judgment-based behaviors seen in real user operations, which highlights the key design principles detailed in Section 3.

## 3 GAMBIT

### 3.1 TASK FORMULATION

The **GAMBIT** dataset consists of a set of $k$ mobile GUI task episodes $\mathcal{T} = \{T_1, T_2, ..., T_k\}$ generated from $m$ candidate mobile applications $\mathcal{A} = \{A_1, A_2, ..., A_m\}$. As illustrated in Fig 2, for each application $A_i$, we define an *atomic instruction* set $\{(i_n, c_n)|A_i\}_{n=1}^N$, where $i_n$ represents an available instruction for this application and $c_n$ specifies its associated constraint (if applicable). Following prior work Lu et al. (2024); Rawles et al. (2023); Li et al. (2024b), each task $T_j$ is represented as a sequence of GUI interaction episodes:

$$T_j = \{(I, i_t, c_t, g_t, a_t)\}_{t=1}^T, \tag{1}$$

where $I$ denotes the global natural language instruction for the task, $i_t$ is the atomic instruction for step $t$, $c_t$ is the constraint, $g_t$ is the screenshot at the current step, and $a_t$ is the executed GUI action. Notably, an atomic instruction $i_t$ may internally correspond to multiple low-level GUI actions and their associated screenshots, while still being treated as a single high-level instruction in our formulation.

### 3.2 ATOMIC INSTRUCTION AND CONSTRAINT COLLECTION

Previous mobile GUI benchmarks typically scale instructions through template substitution or linearly concatenation, which restricts semantic diversity and overlooks realistic application-specific constraints (e.g., *"share via social media"* is a generic action appearing across apps regardless of their distinctive functionalities). To address these limitations, we design a structured, multi-stage pipeline combining human knowledge, LLMs augmentation and rigorous filtering. As illustrated in Fig 2(a), the pipeline consists of: 1) **Human Seeding**: annotators curate a seed set of core executable atomic instructions for each application $A_i$, reflecting its essential user interactions. 2) **LLM-Augmentation**: to expand beyond the handcrafted set, multiple LLMs (e.g., `GPT-4`, `Claude`, `DeepSeek`) are prompted with the application name, app-store description and seed set to iteratively generate additional atomic instructions aligned with the app's functionality. 3) **Constraint Induction**: for each atomic instruction, LLMs propose up to three candidate constraints grounded in application context. Constraints are categorized into a compact taxonomy, including numeric

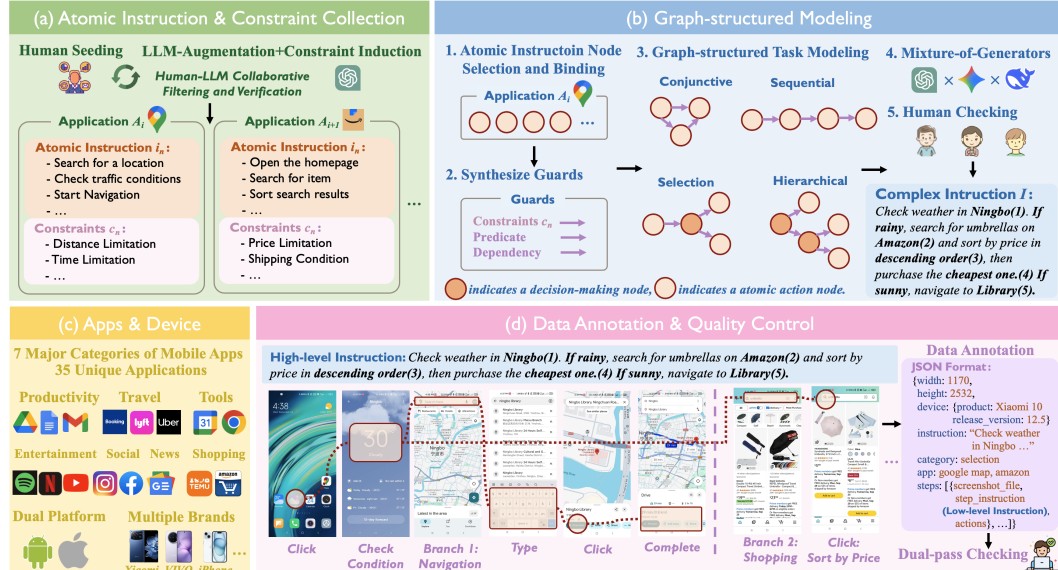

Figure 2: Illustration of GAMBIT construction pipeline.

ranges (e.g., *"price within budget"*), thresholds (e.g., *"rating≥ 4.5"*), boolean attributes (e.g., *"free shipping"*), temporal conditions (e.g., *set the event start time to 3:00 p.m.*) and preference-based filters (e.g., *"find news related to Sports"*). 4) **Filtering and Verification**: automated LLM-based pre-filtering removes infeasible or redundant outputs, followed by human-in-the-loop verification to ensure executability, realism and diversity. Instructions with trivial slot-filling entities or merely superficial lexical variation are rejected (e.g., *"search for Sports news"* v.s. *"find news related to Technology"* are considered duplicates at the atomic instruction level.), and only semantically distinct variations are retained. This pipeline yields a diverse library of atomic instructions enriched with task and application-specific constraints, laying a solid foundation for constructing graph-structured complex tasks in subsequent section.

### 3.3 COMPLEX TASK INSTRUCTION CONSTRUCTION

Building on these atomic instructions, we construct **Graph-structured Complex Tasks** that better approximate real-world user workflows. Each task is modeled as a directed graph $\mathcal{G} = (\mathcal{V}, \mathcal{E})$, where nodes $v \in \mathcal{V}$ are *atomic instructions* and edges $e \in \mathcal{E}$ are annotated with *guards* (constraints, predicates, or dependencies) specifying valid transitions. Each node is linked to the corresponding screenshot $g_t$ at execution step $t$, while guards may reference GUI state (e.g., stock availability, rating) or contextual signals (e.g, time, weather).

**Graph Topologies.** As illustrated in Fig 2(b), we instantiate four canonical graph structures tailored to **atomic action-centric** mobile GUI tasks, extending beyond conventional template-substitution datasets by incorporating **branching and long-horizon reasoning**: 1) **Conjunctive**: multiple goals that must all be satisfied, with flexible ordering unless constrained by guards. 2) **Sequential**: a long linear workflow of atomic instructions. 3) **Conditional**: a task consisting of a binary decision node with two guarded branches (if/else). 4) **Hierarchical**: a multi-level decision tree formed by stacking conditional nodes to capture sophisticated fallback and adaptive behaviors.

**Generation Process.** Given an application $\mathcal{A}_i$ with its atomic instruction set $\{(i_n, c_n)|A_i\}_{n=1}^N$, we first sample a target graph topology $\tau \in \{$Conjunctive, Sequential, Conditional, Hierarchical$\}$ with desired graph length and branching depth. A reasoning LLM Team et al. (2023); Anthropic (2025); Liu et al. (2024a)is then prompted to: **select and bind nodes** by choosing a coherent subset of atomic instructions, **synthesize guards** to edges, and **construct the task** by generating a global natural-language instruction $I$.

**Quality Control.** We observe complementary behaviors across models: `GPT` models Achiam et al. (2023) combine atomic instructions coherently but may generate overly rigid or unrealistic rules (e.g., *"follow a post if likes are odd, else comment"*); `Deepseek` Liu et al. (2024a) produces detailed rationales and explicit restructuring process, but often yields shallow rearrangements lacking

Table 1: Comparison of GUI agent benchmarks

| Dataset | Unique Inst. | Apps | Avg. Steps | High-level & Low-level? | GT | Decision-making? | Platform | Cross-App? |
|---|---|---|---|---|---|---|---|---|
| PixelHelp Li et al. (2020) | 187 | 4 | 4.2 | high-level | ✔ | | Single | |
| MoTIF Burns et al. (2021) | 276 | 125 | 4.5 | high-level & low-level | ✔ | | Single | |
| AITW Rawles et al. (2023) | 30,378 | 357 | 6.5 | high-level | ✔ | | Single | |
| AITZ Zhang et al. (2024b) | 2,504 | 70 | 7.5 | high-level & low-level | ✔ | | Single | |
| AndroidControl Li et al. (2024b) | 15,283 | 833 | 4.8 | high-level & low-level | ✔ | | Single | |
| AMEX Chai et al. (2024) | 2946 | 110 | 12.8 | high-level | ✔ | | Single | |
| GUI Odyssey Lu et al. (2024) | 8,334 | 212 | 15.3 | high-level & low-level | ✔ | | Single | ✔ |
| AppAgent Zhang et al. (2025a) | 45 | 10 | / | high-level | | | Single | |
| MobileAgentBench Wang et al. (2024) | 101 | 10 | / | high-level | | | Single | |
| Meta-GUI Sun et al. (2022) | 1,125 | 11 | 5.3 | high-level | ✔ | | Single | |
| AndroidWorld Rawles et al. (2024) | / | 20 | / | high-level | | | Single | ✔ |
| **Ours** | 830 | 35 | 13.3 | high-level & low-level | ✔ | ✔ | Android & iOS | ✔ |

sequential logic; Gemini Team et al. (2023) generates more consistent sequential and branching workflows with realistic decision rules. To leverage complementary model strengthens, we adopt a **mixture-of-generators strategy**, sampling multiple LLMs to generate complex task instructions. All candidate tasks then undergo LLM-based cross-checking for feasibility, coherence, and realism, followed by double-pass human verification to prune trivial variations and ensure semantic diversity. This pipeline yields a library of decision-aware tasks that reflect realistic and adaptive mobile GUI interaction patterns under dynamic conditions, forming the core of GAMBIT and enabling a fine-grained evaluation in Section 4.

### 3.4 DATA ANNOTATION AND QUALITY CONTROL

**Setup.** We employed 20 professional annotators to label the complex mobile GUI task instructions. Each annotator was provided with the complete **high-level** instruction $I$, the constructed complex task instruction, and asked to decompose it into step-by-step actions. For each execution step, annotators wrote a corresponding **low-level** (step-level) instruction, i.e., a natural-language description of the specific action aligned with the current screenshot. Following the protocol of prior work Li et al. (2024b); Lu et al. (2024), this schema enables us to separately evaluate agent performance at both high-level and low-level granularity. Annotation was performed on Android and iOS devices using real phones. A customized annotation tool recorded interaction data and exported outputs into a unified JSON schema.

**Annotation.** For each step in a task episode, annotators provided the screenshot, low-level (step-level) instruction, action type, and action parameters. Action types were restricted to a predefined set: {Click, Scroll, Type, Navigate to Home, Navigate to Previous Page, Long Press, Complete}. Metadata such as episode IDs, device information, and screenshot dimensions were automatically logged, more details are in Appendix D.

**Quality Control.** To ensure realism, annotators could flag a task as "IMPOSSIBLE" if it violated application usage conventions or could not be executed as instructed. After annotation, the dataset underwent a three-stage quality review: two independent proofreaders and co-authors cross-checked all entries to eliminate residual errors and confirm task executability.

### 3.5 DATASET STATISTICS

We summarize the statistics of GAMBIT in Tab 1. The dataset contains 830 complex task episodes spanning 11,345 actions steps, covering 35 mainstream mobile applications across 7 major categories (Productivity, Travel, Tools, Entertainment, Social Networking, News, Shopping and Payment). For instruction complexity, the average high-level instruction length is 32.54 tokens, which is 18.39% longer than previous commonly used dataset Chai et al. (2024). Each task contains on average 3.2 atomic actions and 4 constraints, while the average task depth is 13.3 steps. For graph topology diversity, GAMBIT instantiates four graph workflow topologies: Conjunctive (24.3%), Sequential (33.4%), Conditional (24.0%), and Hierarchical (18.3%). In addition, 12.5% of tasks are cross-app, further emphasizing task complexity. We also include 250 single atomic instruction tasks as an ablation control group to assess dataset quality and difficulty relative to prior benchmarks. For platform coverage, dataset comprises 881 tasks on Android (version 11 to 15) and 199 tasks on iOS (version 18.5), ensuring multi-platform generalization. In total, 14.1% of tasks are available on both systems, while the remainder are platform-specific. Data collection spans 8 phone brands and 16 device models, further enhancing the robustness of GAMBIT.

Tab 1 provides a detailed comparison with prior datasets and benchmarks, demonstrating the significant strengths of GAMBIT in instruction length, branching decision making structures, cross-app coverage, platform diversity, and dual-level annotation (high-level and low-level), thereby establishing a more challenging and realistic benchmark for evaluating mobile GUI agents.

## 3.6 Evaluation Metric

Prior mobile GUI benchmarks typically adopt metrics such as **Exact Match** (EM), **Type Match** (TM), **Success Rate** (SR) and **Goal Progress** (GP). EM  Li et al. (2024b) requires both agent's predicted action type and parameters to match the ground truth, while TM only checks the action type. SR measures whether all steps in an episode are executed correctly, but collapses into a binary outcome that is often very low for long instructions and fails to indicate which specific actions caused failure. GP measures the fraction of consecutive correct steps from the beginning of the task, but cannot capture branching structures.

As a result, existing evaluation protocols fail to capture two key aspects: **task length sensitivity**, where longer and more complex workflows should contribute more heavily to overall performance, and **decision accuracy in branching structures**, which is essential for realistic mobile task execution. To address these gaps, we introduce two complementary metrics: 1. **Weighted Longest Common Subsequence (W-LCS)**: $W\text{-}LCS = \sum_{i \in \text{LCS}} w_i$, where $w_i = \frac{i}{Length(\mathcal{T}^*)}$. For a given predicted task sequence $\hat{\mathcal{T}}$ and gold sequence $\mathcal{T}^*$, we compute the longest common subsequence with task length-dependent weights. This assigns higher weights to longer decision branches, emphasizing correctness in long-horizon planning. We also compute **Decision Accuracy** and discuss in Section 4.3. For tasks represented as graphs, we evaluate the accuracy of branch decisions at each conditional edge.

## 4 Experiments

### 4.1 Experiment Setting

We comprehensively evaluate existing GUI agents on GAMBIT. The evaluated agents span three categories: 1) **General-purpose GUI agents** trained on a mixture of desktop, web and mobile environments, including AGUVIS-7B Xu et al. (2024), UI-TARS-7B Qin et al. (2025), Qwen2.5-VL-7B Bai et al. (2025), OS-Atlas-Pro-7B Wu et al. (2024); 2) **Mobile-specialized agents** optimized for mobile GUI interactions, AgentCPM-GUI-8B Zhang et al. (2025c); 3) **Reasoning agents** incorporating explicit reasoning process, InfiGUI-R1-3B and InfiGUI-R1-3B (thinking) Liu et al. (2025b). Following established evaluation protocols Zhang et al. (2025c); Lu et al. (2024), each agent is provided with the global high-level instruction $I$, low-level (step-level) instruction $i_t$, constraint $c_t$ (if applicable) and corresponding screenshot $g_t$ at each step $t$. Experiments are conducted according to each agent's official implementation for fairness and reproducibility. Since the action type sets supported by each agent vary, we performed a unified mapping to our predefined set in Section 3.4 to ensure consistent evaluation.

### 4.2 Main Results

Tab 2 reports the overall performance of evaluated agents on GAMBIT, we observed several key findings. Our benchmark contains a difficulty gradient and shows as a clear SR and GP degrade monotonically with topology complexity increases. On the most complex Hierarchical subset, the average SR falls to 14.00% even under low-level instructions, and GP averages merely 13.10% under high-level instructions. Among all agents, AGUVIS-7B achieves the highest 89.81% overall EM with low-level setting and AgentCPM-GUI-7B with the highest 53.31% EM with high-level setting. InfiGUI-R1-3B-thinking shows the smallest degradation (2.38%) from Conjunctive/Sequential to Conditional/Hierarchical, suggesting strongest robustness to length and structural variation. These agent's top performance highlights a wider coverage of task and more robust to structural variation in their training.

**Comparison with prior benchmarks.** To ensure the difficulty of GAMBIT is not inflated by data collection (e.g., artificially extended step descriptions or redundant GUI elements), we include a

Table 2: Main results for mobile GUI agents. "InfiGUI-R1-3B tk" indicates the thinking mode for InfiGUI-R1-3B, and this notation remains in following tables.

| Model | Level | Single | | | | Conjunctive | | | | Sequential | | | | Conditional | | | | Hierarchical | | | |
|---|---|---|---|---|---|---|---|---|---|---|---|---|---|---|---|---|---|---|---|---|---|
| | | EM | TM | SR | GP | EM | TM | SR | GP | EM | TM | SR | GP | EM | TM | SR | GP | EM | TM | SR | GP |
| AGUVIS-7B | LL | 87.81 | 95.55 | 72.40 | 72.47 | 88.75 | 97.56 | 50.00 | 61.39 | 90.70 | 97.87 | 41.88 | 59.55 | 90.51 | 97.07 | 49.25 | 72.46 | 91.30 | 98.32 | 33.55 | 67.18 |
| | HL | 42.59 | 59.44 | 32.00 | 26.91 | 33.33 | 60.01 | 3.48 | 11.75 | 26.71 | 49.18 | 0.36 | 11.86 | 22.49 | 41.27 | 0.50 | 16.83 | 21.54 | 42.38 | 0.66 | 12.42 |
| UI-TARS-7B | LL | 87.11 | 97.20 | 63.60 | 70.03 | 79.51 | 97.37 | 26.24 | 44.61 | 80.72 | 96.62 | 24.55 | 43.76 | 85.33 | 96.18 | 26.13 | 59.11 | 83.36 | 97.51 | 15.79 | 49.79 |
| | HL | 63.34 | 79.56 | 31.60 | 35.68 | 53.97 | 78.25 | 3.96 | 14.49 | 55.03 | 79.31 | 2.53 | 16.48 | 48.63 | 69.64 | 1.01 | 18.71 | 45.93 | 71.87 | 0.66 | 15.39 |
| Qwen2.5-VL-7B | LL | 79.90 | 91.21 | 62.80 | 59.99 | 81.26 | 92.98 | 36.14 | 51.76 | 76.62 | 87.94 | 24.19 | 46.25 | 80.44 | 91.65 | 19.10 | 51.61 | 75.68 | 86.08 | 11.18 | 48.32 |
| | HL | 58.55 | 73.53 | 38.00 | 35.66 | 53.66 | 71.43 | 5.45 | 17.45 | 44.80 | 59.01 | 4.33 | 18.70 | 41.82 | 57.53 | 1.01 | 19.54 | 36.31 | 50.99 | 0.66 | 16.02 |
| OS-Atlas-Pro-7B | LL | 80.99 | 90.94 | 67.60 | 67.32 | 74.94 | 90.02 | 31.19 | 47.03 | 67.10 | 83.76 | 18.05 | 39.53 | 68.51 | 82.75 | 16.58 | 46.34 | 60.98 | 79.74 | 4.61 | 36.78 |
| | HL | 54.31 | 70.38 | 36.40 | 33.12 | 46.21 | 73.38 | 6.44 | 16.25 | 47.34 | 72.09 | 1.44 | 12.95 | 40.03 | 65.04 | 0.00 | 14.10 | 37.57 | 65.60 | 0.00 | 10.96 |
| AgentCPM-GUI-8B | LL | 83.85 | 90.14 | 68.80 | 69.05 | 84.20 | 93.26 | 41.09 | 54.09 | 86.77 | 93.91 | 28.16 | 49.82 | 86.95 | 92.42 | 33.17 | 61.06 | 87.74 | 93.26 | 19.08 | 57.58 |
| | HL | 62.48 | 75.82 | 32.40 | 35.73 | 57.94 | 78.14 | 8.91 | 21.53 | 55.97 | 78.84 | 2.53 | 16.90 | 50.39 | 71.23 | 1.01 | 18.15 | 47.28 | 73.03 | 0.66 | 15.00 |
| InfiGUI-R1-3B | LL | 58.26 | 71.98 | 59.60 | 46.93 | 61.48 | 75.09 | 25.74 | 38.57 | 62.81 | 80.26 | 13.00 | 35.58 | 57.87 | 74.71 | 9.55 | 38.29 | 59.19 | 77.81 | 3.95 | 32.00 |
| | HL | 37.58 | 50.63 | 27.20 | 24.11 | 38.03 | 57.39 | 0.99 | 10.19 | 42.73 | 63.38 | 1.08 | 8.91 | 37.07 | 56.30 | 0.00 | 10.49 | 38.54 | 62.58 | 0.66 | 9.75 |
| InfiGUI-R1-3B tk | LL | 60.97 | 74.11 | 65.60 | 49.28 | 69.78 | 84.74 | 34.65 | 44.29 | 69.77 | 89.80 | 22.74 | 43.14 | 63.57 | 83.80 | 17.09 | 44.20 | 66.53 | 88.53 | 9.87 | 41.52 |
| | HL | 44.35 | 63.29 | 35.60 | 28.77 | 44.99 | 70.57 | 4.95 | 14.08 | 49.06 | 73.32 | 1.81 | 12.59 | 40.28 | 63.82 | 2.01 | 16.21 | 41.97 | 68.08 | 0.00 | 12.22 |

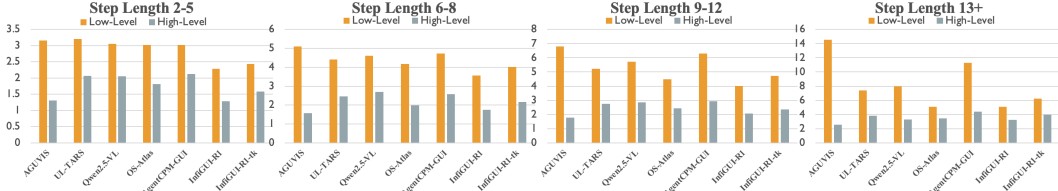

Figure 4: Visualization of W-LCS scores as a function of task step length.

Single Atomic Instruction subset as control group. Agent performance on this subset is comparable to that on prior single-step benchmarks (81.19% on ours v.s. 80.81% on AndroidControl Li et al. (2024b), a widely used mobile GUI benchmark), confirming that GAMBIT preserves the feasibility of basic action execution.

To isolate dynamic differences beyond Section 3.5, we compare the agents performance between GAMBIT and widely adopted mobile GUI datasets. Averaged across instruction topologies, GAMBIT induces an accuracy drop of 20.69% relative to prior datasets. Unlike AndroidControl and related sequential-only benchmarks, our dataset penalizes structural errors more severely: under high-level instructions, EM is 24.69% lower than AndroidControl, and even AndroidControl's hardest split remains 26.29% higher than our Conditional/Hierarchical subsets. These results highlight that GAMBIT introduces **decision-aware difficulty** absent from earlier datasets, while still maintaining parity on atomic actions. It provides a finer-grained diagnostic lens for distinguishing agent capabilities under realistic long-horizon and branching workflows.

## 4.3 ABLATION STUDIES

**High-level v.s. Low-level Instructions.** Tab 2 contrasts agents under low-level (step-specified) and high-level (goal-only) instructions. Across all models, EM/TM are consistently higher with low-level guidance, confirming that most agents can reliably execute explicitly grounded atomic actions. When shifted to high-level goals, TM drops by 24.11% on average and EM falls below 32.78%. This exposes a persistent gap between current agents' strong visual grounding/navigation and weak global task reasoning. In real-world scenarios, users rarely issue perfectly disambiguated step instructions and this gap becomes relevant as task complexity increases.

**Impact of Task Length and Branching.**

Under low-level settings (Tab. 2), TM/EM remains relatively insensitive to task length, as each step is explicitly grounded and largely independent. In contrast, high-level instructions show strong length sensitivity (Fig. 3): GP decreases sharply as sequence length increases, reflecting cumulative error propagation across steps. Once sequences exceed 6–8 steps, GP drops below 20% for all models, indicating that current agents struggle to sustain long-horizon reasoning and execution beyond short workflows.

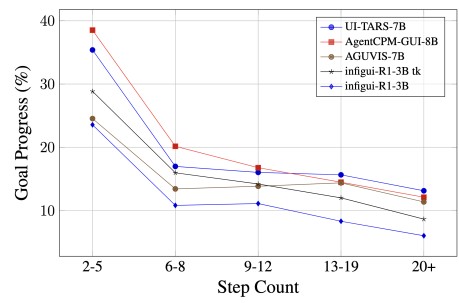

Figure 3: High-level GP v.s. Step Length.

Table 3: Decision accuracy (EM ↑) at different branching depths under high-level instructions. "–" denotes accuracy degradation on deeper nodes.

| Model | First Decision | Deeper Decisions | EM Difference |
|---|---|---|---|
| AGUVIS-7B | 31.29% | 29.11% | -2.18% |
| UI-TARS-7B | 45.85% | 29.72% | -16.13% |
| Qwen2.5-VL-7B | 37.46% | 30.56% | -6.90% |
| OS-Atlas-Pro-7B | 32.38% | 29.72% | -2.66% |
| AgentCPM-GUI-8B | 37.25% | 29.72% | -7.53% |
| InfiGUI-R1-3B | 40.17% | 37.95% | -2.22% |
| **InfiGUI-R1-3B tk** | **41.31%** | **41.83%** | **+0.52%** |

To better capture partial progress and robustness in long-horizon tasks, we report W-LCS, which quantifies how far an agent proceeds before failure, aligning with real user tolerance where partial completion still has value. As shown in Fig 4, `AgentCPM-GUI-8B` emerges as the strongest performer as task length increases, even though its GP and SR are not the highest in Tab 2. `InfiGUI-R1-3B (thinking)` exhibits only moderate W-LCS under low-level settings, but its advantage becomes pronounced on tasks exceeding 13 steps with high-level instructions, ranking among the top-2 across all agents. This suggests their superior resistance to error accumulation and greater robustness in complex reasoning. In contrast, `AGUVIS-7B` shows the opposite trend: strong W-LCS under low-level tasks but severe degradation on high-level ones, indicating strong grounding ability but limited capacity for global task planning.

**Branching structures amplify difficulty.** Conditional/Hierarchical tasks introduce decision nodes where one misjudgment invalidates entire subtrees, leading to sharper declines than Sequential/Conjunctive workflows. To quantify this, we measure decision accuracy by branch depth in Tab 3. The first decision in Conditional and Hierarchical graphs averages only 36.85%, while deeper branches in Hierarchical tasks perform worse by an additional 7.08% dropping on average. Notably, the `InfiGUI-R1-3B` series, a smaller model series with reasoning SFT and RL, outperforms other agents by 3.89% on first decision and 10.12% on deeper nodes, maintaining stable accuracy as depth increases. These findings indicate that **reasoning-oriented training enhances decision robustness**, a property largely absent in baseline agents. These results underscore that deeper and more complex branching makes models less reliable, exposing real-world challenges that linear workflows in prior datasets cannot reveal.

## 4.4 CASE STUDIES AND ERROR ANALYSIS

**Q1. Why agents fail at decision-making steps?** We analyze decision-aware failure cases using `InfiGUI-R1-3B`, the only agent that exposes intermediate reasoning traces. As shown in Fig 5, two recurrent failure modes emerge: *Mode 1: Ignoring branch constraints.* The agent often defaults to the first available option instead of following the explicit branching condition, thereby bypassing the decision step. *Mode 2: Plausible but false reasoning.* The agent produces a coherent chain-of-thought that acknowledges the instruction and GUI state, but then executes an action inconsistent with its own reasoning, again defaulting to the first option. These patterns reveal a clear gap: agents are proficient at single-step execution (Tab 2 shows near-saturated TM/EM on Single, short Sequential workflows and low-level settings), yet struggle with conditional reasoning, where multiple candidate actions must be weighted against global instructions and GUI context. A key reason is that **existing training corpora for mobile GUI agents lack decision-oriented trajectories**, leaving models underexposed to branching supervision. GAMBIT addresses this gap by explicitly incorporating complex graph topologies and systematic evaluation of decision accuracy.

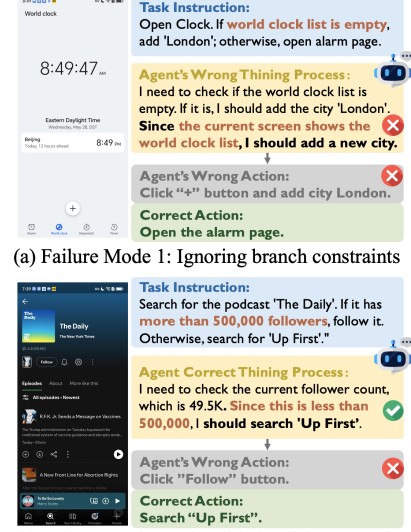

(a) Failure Mode 1: Ignoring branch constraints

(b) Failure Mode 2: Plausible but false reasoning

Figure 5: Decision failure cases.

**Q2. Beyond task length and branching, what other factors limit agent performance?**

We identify three additional factors that consistently affect model robustness.

Table 4: Model performance on different action types.

| Model | CLICK | | TYPE | | SCROLL | | Complete | | Others | |
|---|---|---|---|---|---|---|---|---|---|---|
| | TM | EM | TM | EM | TM | EM | TM | EM | TM | EM |
| **Low-level** | | | | | | | | | | |
| AGUVIS-7B | 99.84 | 91.65 | 95.28 | 91.89 | 99.04 | 87.33 | 86.85 | 86.85 | 93.69 | 83.95 |
| UI-TARS-7B | 99.16 | 79.01 | 94.40 | 89.15 | 99.00 | 94.19 | 99.04 | 99.04 | 44.97 | 44.97 |
| Qwen2.5-VL-7B | 93.12 | 83.21 | 84.34 | 81.49 | 81.01 | 46.04 | 89.86 | 89.86 | 55.38 | 53.23 |
| OS-Otlas-Pro-7B | 94.96 | 74.66 | 85.67 | 82.16 | 48.65 | 28.94 | 67.31 | 67.31 | 68.51 | 62.66 |
| AgentCPM-GUI-8B | 97.27 | 88.30 | 89.32 | 84.98 | 98.27 | 96.61 | 79.79 | 79.79 | 40.75 | 31.51 |
| InfiGUI-R1-3B | 87.04 | 79.14 | 85.16 | 82.67 | 81.81 | 5.64 | 10.43 | 10.43 | 86.17 | 22.95 |
| InfiGUI-R1-3B tk | 97.94 | 88.79 | 89.78 | 86.84 | 96.28 | 7.69 | 8.91 | 8.91 | 90.56 | 24.21 |
| **High-level** | | | | | | | | | | |
| AGUVIS-7B | 67.48 | 34.68 | 28.69 | 23.52 | 14.27 | 11.02 | 0.00 | 0.00 | 43.83 | 43.83 |
| UI-TARS-7B | 88.45 | 54.74 | 64.32 | 53.35 | 52.34 | 47.74 | 44.81 | 44.81 | 28.53 | 28.53 |
| Qwen2.5-VL-7B | 67.41 | 44.79 | 53.25 | 48.00 | 15.25 | 11.40 | 79.66 | 79.66 | 18.87 | 18.87 |
| OS-Otlas-Pro-7B | 86.37 | 48.53 | 54.50 | 44.43 | 29.85 | 26.71 | 35.55 | 35.55 | 41.61 | 41.61 |
| AgentCPM-GUI-8B | 86.50 | 54.24 | 64.67 | 52.45 | 45.42 | 41.83 | 66.19 | 66.19 | 29.56 | 29.24 |
| InfiGUI-R1-3B | 88.56 | 55.16 | 57.24 | 49.33 | 41.38 | 23.45 | 9.05 | 8.98 | 36.48 | 29.87 |
| InfiGUI-R1-3B tk | 75.17 | 46.57 | 54.22 | 47.91 | 32.42 | 20.24 | 17.47 | 17.40 | 33.65 | 28.00 |

**Action Type Effects.** Tab 4 highlights marked variation across action types. `Long Press`, a realistic yet underrepresented action, shows the lowest accuracy across models (on average 66.31% lower than `Click`/`Type`). For UI-TARS and OS-Atlas, the `Complete` action underperforms other models by 52.03%, suggesting their difficulty in detecting task termination under long-horizon goals. Their overall SR rises by 8.18% when `Complete` is excluded, indicating that many errors reflect termination misrecognition rather than execution failures. Annotators also noted the near absence of `Double-Tap`: although often replaceable by alternative actions, its omission from most agents' action space and training data may limit generalization. These results indicate that action space diversity and explicit modeling of final-state inference remain a performance improvement direction.

**History Window Length.** Long-horizon tasks impose higher demands on both historical context and efficiency. In mainstream settings, history length varies from each agent. For instance, `AgentCPM` achieves the best high-level accuracy with a 4-step window (1.38% higher than full-history or no-history), whereas its low-level accuracy declines as history grows, indicating that longer context is not uniformly beneficial. For `AGUVIS`, it performs better with longer history under high-level instructions, but shows little differences for low-level setting. We identify this as a trade-off among multiple factors: 1) **Contextual Overload**: excessively long textual/ histories may increase confusion and accumulative error carry-over. 2) **Visual Down-sampling**: aggregating many screenshots reduces effective resolution, harming precision actions such as small UI target clicks. 3) **Efficiency**: longer histories increase inference latency, problematic for real-time mobile scenarios. By contrast, the no-history setting runs faster inference but underperforms due to insufficient grounding (1.15% lower EM). Effective history utilization depends on model design rather than merely context length, and saliency-aware or memory-pruned histories are future directions.

**Instruction Language and Platform.** We further separate results by instruction language and operation systems (more results in Appendix H and G). `Qwen2.5-VL-7B` is the most robust to English-Chinese instruction shifts due to its training corpora distribution, with 3.5% higher EM on Chinese instruction. In addition, we observe that `InfiGUI-R1-3B` series benefits from its "thinking" prompting, effectively reduces outputs formatting errors. For all tested agents, high-level instructions produce more irregular responses than low-level ones, and Chinese prompts are more error-prone than English. As for platform difference, since most mobile GUI training corpora skewed toward Android, average performance drops by 5.30% on iOS, likely due to GUI styling and action differences. These factors, spanning action coverage, history utilization, multilingual grounding, and multi-platform robustness, are orthogonal to the challenges discussed in Section 4.3. GAMBIT systematically exposes both decision-aware reasoning failures and broader robustness limitations, providing a rigorous and comprehensive benchmark for future mobile GUI agents.

## 5 CONCLUSION

In this paper, we introduced **GAMBIT**, the first benchmark targeting *long-horizon workflows* and *decision-aware execution* in mobile GUI agents. Unlike prior template-based datasets, it captures realistic user interactions through graph-structured instructions and cross-platform annotation. Experiments show that while current agents handle single-step execution well, they struggle with long chains, branching, and generalization, making GAMBIT a comprehensive testbed for mobile agents.

# 6 ETHICS STATEMENT

GAMBIT was constructed with careful consideration of privacy, safety and fairness. All task instructions and screenshots were collected by professional annotators on mobile devices using publicly available mobile applications and no personal or sensitive user data are included. The dataset contains only synthetic task instructions generated during controlled annotation as stated in Section 3, ensuring that no private information is exposed. No emulators are used for annotation.

All annotators, co-authors and quality checkers are clearly notified the instruction, meta data and screenshot usage. Annotators are paid at market price according to local laws and requirements.

All applications were operated in safe environments without accessing sensitive content such as payments, contacts, or personal communications(all mentioned user profiles, names, emails and accounts are separately created with no relations to real-world individuals). The dataset is intended solely for academic research on mobile GUI agents and is released under a non-commercial license to discourage misuse.

# 7 REPRODUCIBILITY STATEMENT

Our paper's main contribution is a benchmark dataset, currently it is available in anonymous code base for review: `https://anonymous.4open.science/r/GAMBIT-40BB/`. We would open-source all instructions, annotated data, screenshots, metadata and sufficient experimental code (inference and evaluation) for reproducibility. The construction pipeline code, hyperparameters, settings and prompt swill also be included in open-source code base, including sufficient guidelines. Our current implementation are under each agents' official guidance and settings, detailed in code base.

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

## A  APPENDIX: USE OF LLMS

Our LLM usage are stated in Section 3 and concluded as below. *Instruction Augmentation*: we utilized multiple LLMs for expanding human seed atomic instructions, collecting constraints and iteratively refinement with human collaboration. *Complex Instruction Construction*: we utilized multiple LLMs for construction longer instructions from atomic actions, iterative enhancing and quality checking (e.g., grammar, coherence, feasibility). We ensured that all LLM generated instructions are double verified by either paper co-authors or annotators to prohibit unsage/inapproprate instructions, with essential manual re-writing.

The experiments related to mobile GUI agents are conducted with LLMs, all models (checkpoints, configurations and licenses) are obtained from their official open-source implementation without additional training or modification. We did not use LLM-generated contents for paper writing, but merely grammar checking.

## B  APPENDIX: LIMITATION

We acknowledge that although GAMBIT covers diverse application categories, platforms (Android/iOS), and bilingual instructions (English/Chinese), it remains limited in cultural and linguistic scope, including more device brands and models (e.g., Mobile Tablets and iPads), more application categories and comparison of same application under different langue settings (e.g., Chinese applications may contain advertisements compared with their English versions). Future extensions may address these limitations to ensure broader fairness and inclusiveness. We consider actively incorporating more downstream application categories such as Medical, Education and Games.

We also acknowledge that current workflows are human-LLM approximation of real-world user actions, and may not fully reflect all user cases and complex decisions. Additionally, although we allow annotators mark some tasks as "IMPOSSIBLE", ambiguous or conflicting tasks may still exist in real user requests and our dataset has limited coverage for such scenarios.

Despite employing 20 professional annotators and a dual-review process, a small number of ambiguous or unclear low-level instructions may still exist. The linguistic diversity of high-level instructions remains limited (ambiguous, colloquial phrasing characteristic of real users), with most maintaining relatively standardized expressions.

We did not systematically verify potential overlap between GAMBIT and other agents' training data in this work (although our data are unique by paper submission), so we cannot entirely rule out the possibility that a small number of task patterns appeared in other agents' training.

## C  APPENDIX: PROMPT FOR INSTRUCTION GENERATION

We provide detailed prompt templates for utilizing LLMs for generating complex instructions. The prompts are available in our code base `https://anonymous.4open.science/r/GAMBIT-40BB/`.

# D    APPENDIX: ANNOTATION DETAILS

We wrote a standardized annotation guideline for the 20 annotators, with example JSON file format, example annotated data and other required information. Here we provide a snapshot of our proposed dataset meta data, with more details available in code base.

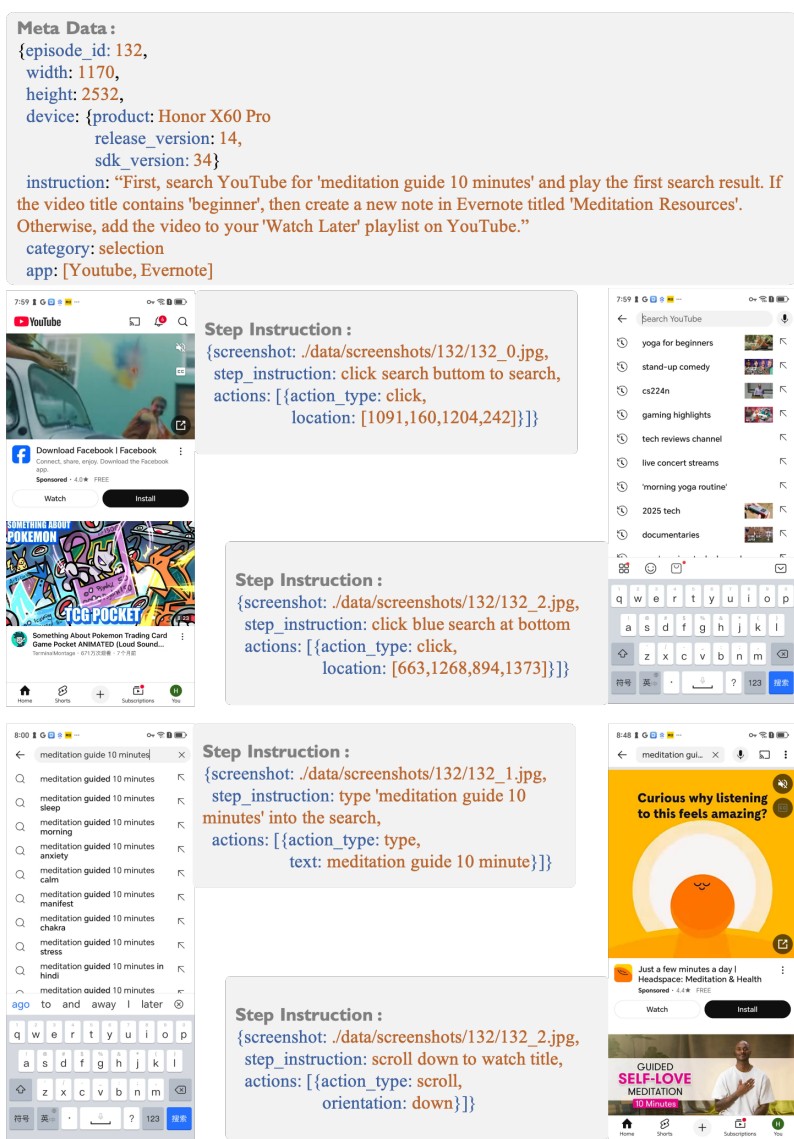

Figure 6: Illustration of annotation details and meta data.

# E  APPENDIX: STEP LENGTH EXPERIMENTS

Here we provide detailed experimental Results of Agent's performance across different step length in compensation to Section 4.3 discussions. Here 1a denotes atomic action level GP, 1b denotes atomic action level GP with branching length weighted. 2a denotes step level GP, 1b denotes step level GP with branching length weighted. 3a denotes the longest common sequence from first step, 3b denotes the longest common sequence from first step with branching length weighted. 4a denotes the longest common sequence taking all steps into consideration, 4b denotes the longest common sequence taking all steps into consideration with branching length weighted. 5a denotes the longest common sequence taking all steps into consideration with task length weighted, 5b denotes the longest common sequence taking all steps into consideration with branching length and task length weighted. 6a denotes the longest common sequence percentage, 6b denotes the longest common sequence percentage with task length weighted.

Table 5: Model performance(GP) on different step size

| Model | Group | Level | Tasks | 1a (%) | 1b (%) | 2a (%) | 2b (%) | 3a (Abs) | 3b (Abs) | 4a (Abs) | 4b (Abs) | 5a (Abs) | 5b (Abs) | 6a (%) | 6b (%) |
|---|---|---|---|---|---|---|---|---|---|---|---|---|---|---|---|
| **AGUVIS** | **2-5 steps** | HL | 281 | 1.48 | 2.76 | 0.85 | 0.95 | 0.03 | 0.06 | 1.20 | 1.31 | 0.81 | 0.84 | 24.54 | 22.63 |
| | | LL | 281 | 63.32 | 62.72 | 62.56 | 61.28 | 0.75 | 0.89 | 2.90 | 3.16 | 2.55 | 2.75 | 72.53 | 71.39 |
| | **6-8 steps** | HL | 225 | 6.35 | 8.82 | 4.12 | 4.17 | 0.17 | 0.24 | 1.58 | 1.58 | 0.85 | 0.84 | 13.43 | 13.11 |
| | | LL | 226 | 54.03 | 58.12 | 52.00 | 52.12 | 1.15 | 1.51 | 5.02 | 5.09 | 3.96 | 4.02 | 62.02 | 61.91 |
| | **9-12 steps** | HL | 222 | 10.79 | 12.32 | 6.27 | 6.07 | 0.28 | 0.34 | 1.77 | 1.79 | 1.13 | 1.14 | 13.85 | 13.58 |
| | | LL | 222 | 56.73 | 59.54 | 53.94 | 53.57 | 1.42 | 1.71 | 6.67 | 6.79 | 5.49 | 5.58 | 64.77 | 64.51 |
| | **13+ steps** | HL | 351 | 10.49 | 10.99 | 5.94 | 5.16 | 0.33 | 0.38 | 2.51 | 2.57 | 1.76 | 1.80 | 13.16 | 11.80 |
| | | LL | 351 | 59.32 | 59.82 | 55.16 | 55.05 | 1.91 | 2.14 | 11.98 | 14.56 | 9.78 | 11.79 | 65.16 | 64.89 |
| | **Overall** | HL | 1079 | 7.34 | 9.94 | 4.30 | 4.85 | 0.21 | 0.31 | 1.82 | 2.18 | 1.19 | 1.47 | 16.32 | 13.23 |
| | | LL | 1080 | 58.72 | 59.81 | 56.17 | 54.91 | 1.35 | 1.79 | 7.07 | 10.88 | 5.80 | 8.82 | 66.34 | 64.98 |
| **AgentCPM** | **2-5 steps** | HL | 281 | 28.20 | 27.40 | 27.92 | 23.97 | 0.33 | 0.37 | 1.99 | 2.12 | 1.25 | 1.27 | 38.52 | 34.99 |
| | | LL | 281 | 53.05 | 55.01 | 52.57 | 51.07 | 0.65 | 0.81 | 2.77 | 3.02 | 2.44 | 2.64 | 69.79 | 68.59 |
| | **6-8 steps** | HL | 226 | 13.09 | 16.46 | 11.42 | 11.42 | 0.32 | 0.46 | 2.56 | 2.58 | 1.27 | 1.27 | 20.16 | 19.83 |
| | | LL | 226 | 50.21 | 54.25 | 48.10 | 47.65 | 1.08 | 1.42 | 4.66 | 4.72 | 3.73 | 3.75 | 58.90 | 58.39 |
| | **9-12 steps** | HL | 222 | 12.57 | 15.37 | 8.95 | 8.56 | 0.34 | 0.45 | 2.91 | 2.94 | 1.37 | 1.37 | 16.78 | 16.29 |
| | | LL | 222 | 50.90 | 56.91 | 47.13 | 46.87 | 1.32 | 1.67 | 6.19 | 6.30 | 4.78 | 4.88 | 57.45 | 57.36 |
| | **13+ steps** | HL | 351 | 10.85 | 11.49 | 6.62 | 5.92 | 0.37 | 0.42 | 4.01 | 4.40 | 1.89 | 2.03 | 13.52 | 12.37 |
| | | LL | 351 | 45.50 | 47.52 | 39.48 | 38.77 | 1.52 | 1.75 | 9.49 | 11.31 | 7.22 | 8.64 | 49.05 | 48.28 |
| | **Overall** | HL | 1080 | 16.19 | 15.06 | 13.65 | 8.67 | 0.34 | 0.43 | 2.96 | 3.69 | 1.49 | 1.74 | 22.09 | 16.00 |
| | | LL | 1080 | 49.56 | 51.60 | 46.26 | 42.52 | 1.16 | 1.56 | 6.05 | 8.79 | 4.75 | 6.78 | 58.23 | 53.05 |
| **UI-TARS** | **2-5 steps** | HL | 281 | 25.86 | 24.85 | 25.28 | 20.95 | 0.30 | 0.34 | 1.94 | 2.07 | 1.14 | 1.14 | 35.39 | 31.66 |
| | | LL | 281 | 69.04 | 69.48 | 68.50 | 64.83 | 0.83 | 1.02 | 2.98 | 3.21 | 2.50 | 2.65 | 72.95 | 70.05 |
| | **6-8 steps** | HL | 226 | 9.28 | 12.19 | 7.28 | 7.46 | 0.23 | 0.33 | 2.43 | 2.45 | 1.08 | 1.10 | 16.98 | 16.92 |
| | | LL | 226 | 43.74 | 49.78 | 41.75 | 41.28 | 0.97 | 1.27 | 4.37 | 4.41 | 3.51 | 3.52 | 56.16 | 55.55 |
| | **9-12 steps** | HL | 222 | 12.08 | 14.39 | 7.77 | 7.43 | 0.33 | 0.42 | 2.73 | 2.76 | 1.32 | 1.33 | 16.03 | 15.73 |
| | | LL | 222 | 43.06 | 49.36 | 38.91 | 38.33 | 1.10 | 1.38 | 5.14 | 5.21 | 4.20 | 4.27 | 52.47 | 51.99 |
| | **13+ steps** | HL | 351 | 11.99 | 12.53 | 7.06 | 6.22 | 0.40 | 0.44 | 3.53 | 3.80 | 2.02 | 2.23 | 14.63 | 13.46 |
| | | LL | 351 | 32.41 | 34.10 | 25.44 | 21.37 | 1.05 | 1.20 | 6.86 | 7.39 | 4.86 | 5.05 | 37.42 | 33.44 |
| | **Overall** | HL | 1080 | 15.05 | 14.33 | 12.00 | 7.86 | 0.32 | 0.41 | 2.72 | 3.28 | 1.45 | 1.82 | 20.81 | 15.88 |
| | | LL | 1080 | 46.50 | 44.27 | 42.82 | 30.88 | 0.99 | 1.23 | 4.98 | 6.23 | 3.83 | 4.50 | 53.68 | 42.97 |
| **Qwen2.5-VL** | **2-5 steps** | HL | 281 | 22.21 | 24.35 | 21.55 | 20.89 | 0.29 | 0.40 | 1.88 | 2.06 | 1.24 | 1.28 | 37.15 | 34.75 |
| | | LL | 281 | 51.63 | 53.12 | 50.85 | 52.93 | 0.63 | 0.80 | 2.75 | 3.05 | 2.30 | 2.54 | 63.86 | 64.52 |
| | **6-8 steps** | HL | 226 | 15.10 | 19.33 | 12.37 | 12.49 | 0.39 | 0.56 | 2.68 | 2.70 | 1.37 | 1.39 | 21.04 | 21.03 |
| | | LL | 226 | 45.06 | 50.03 | 43.25 | 43.41 | 1.01 | 1.36 | 4.55 | 4.61 | 3.48 | 3.53 | 53.40 | 53.40 |
| | **9-12 steps** | HL | 222 | 12.16 | 13.87 | 7.90 | 7.48 | 0.32 | 0.38 | 2.83 | 2.85 | 1.26 | 1.26 | 15.56 | 15.17 |
| | | LL | 222 | 45.56 | 49.67 | 42.14 | 41.30 | 1.15 | 1.43 | 5.64 | 5.72 | 4.39 | 4.43 | 53.31 | 52.69 |
| | **13+ steps** | HL | 351 | 12.88 | 13.68 | 8.30 | 7.32 | 0.43 | 0.49 | 3.23 | 3.31 | 2.02 | 2.10 | 15.08 | 13.54 |
| | | LL | 351 | 35.81 | 37.94 | 28.97 | 26.59 | 1.19 | 1.37 | 7.33 | 8.00 | 5.59 | 6.24 | 39.99 | 37.52 |
| | **Overall** | HL | 1080 | 15.62 | 15.94 | 12.52 | 9.17 | 0.36 | 0.47 | 2.68 | 3.03 | 1.52 | 1.78 | 22.17 | 16.62 |
| | | LL | 1080 | 43.87 | 44.35 | 40.36 | 33.83 | 1.00 | 1.32 | 5.21 | 6.70 | 4.04 | 5.23 | 51.74 | 44.78 |
| **OS-Atlas-Pro** | **2-5 steps** | HL | 281 | 13.49 | 13.44 | 13.11 | 11.46 | 0.16 | 0.19 | 1.66 | 1.81 | 1.08 | 1.12 | 32.82 | 30.08 |
| | | LL | 281 | 52.94 | 54.80 | 52.35 | 51.40 | 0.66 | 0.82 | 2.76 | 3.02 | 2.45 | 2.67 | 69.29 | 68.51 |
| | **6-8 steps** | HL | 226 | 8.85 | 12.19 | 6.78 | 6.73 | 0.23 | 0.35 | 1.98 | 1.99 | 0.97 | 0.97 | 15.51 | 15.19 |
| | | LL | 226 | 41.84 | 47.73 | 38.58 | 38.03 | 0.95 | 1.29 | 4.15 | 4.18 | 3.41 | 3.42 | 53.73 | 53.04 |
| | **9-12 steps** | HL | 222 | 9.61 | 11.82 | 6.66 | 6.50 | 0.28 | 0.36 | 2.41 | 2.44 | 1.13 | 1.13 | 13.30 | 13.03 |
| | | LL | 222 | 34.86 | 39.96 | 29.28 | 28.42 | 0.91 | 1.16 | 4.47 | 4.50 | 3.61 | 3.63 | 44.57 | 43.77 |
| | **13+ steps** | HL | 351 | 9.21 | 9.57 | 5.35 | 4.80 | 0.30 | 0.34 | 3.20 | 3.43 | 1.60 | 1.74 | 11.23 | 10.37 |
| | | LL | 351 | 26.14 | 27.39 | 18.65 | 16.55 | 0.86 | 0.97 | 4.95 | 5.07 | 4.03 | 4.19 | 30.16 | 27.17 |
| | **Overall** | HL | 1080 | 10.33 | 10.96 | 7.94 | 5.94 | 0.25 | 0.33 | 2.38 | 2.91 | 1.24 | 1.47 | 18.17 | 13.17 |
| | | LL | 1080 | 38.19 | 36.83 | 33.77 | 24.57 | 0.84 | 1.05 | 4.11 | 4.67 | 3.40 | 3.85 | 48.23 | 37.21 |
| **InfiGUI** | **2-5 steps** | HL | 281 | 6.45 | 7.67 | 5.80 | 5.49 | 0.09 | 0.13 | 1.19 | 1.29 | 0.77 | 0.78 | 23.56 | 21.47 |
| | | LL | 281 | 9.46 | 16.19 | 8.61 | 9.61 | 0.19 | 0.39 | 2.05 | 2.28 | 1.76 | 1.92 | 48.81 | 49.12 |
| | **6-8 steps** | HL | 226 | 5.86 | 9.34 | 4.04 | 4.34 | 0.18 | 0.30 | 1.74 | 1.75 | 0.70 | 0.72 | 10.82 | 11.01 |
| | | LL | 226 | 24.17 | 34.29 | 20.19 | 21.04 | 0.74 | 1.18 | 3.51 | 3.50 | 2.87 | 2.86 | 43.80 | 43.30 |
| | **9-12 steps** | HL | 222 | 8.24 | 10.37 | 4.83 | 4.79 | 0.23 | 0.31 | 2.09 | 2.08 | 0.90 | 0.88 | 11.11 | 10.78 |
| | | LL | 222 | 26.96 | 35.52 | 19.90 | 20.17 | 0.83 | 1.19 | 3.99 | 4.01 | 3.10 | 3.12 | 38.16 | 38.26 |
| | **13+ steps** | HL | 351 | 6.83 | 7.60 | 3.53 | 3.09 | 0.24 | 0.29 | 2.95 | 3.25 | 0.96 | 0.99 | 7.38 | 6.60 |
| | | LL | 351 | 24.19 | 25.71 | 16.72 | 14.98 | 0.82 | 0.97 | 4.96 | 5.11 | 3.67 | 3.76 | 27.85 | 24.68 |
| | **Overall** | HL | 1080 | 6.82 | 8.48 | 4.50 | 3.74 | 0.19 | 0.28 | 2.06 | 2.71 | 0.84 | 0.92 | 13.08 | 9.03 |
| | | LL | 1080 | 20.92 | 28.24 | 15.99 | 16.27 | 0.64 | 0.99 | 3.70 | 4.50 | 2.89 | 3.39 | 38.76 | 31.30 |
| **InfiGUI tk** | **2-5 steps** | HL | 281 | 5.92 | 8.52 | 5.28 | 5.98 | 0.10 | 0.20 | 1.45 | 1.58 | 0.96 | 0.98 | 28.84 | 26.76 |
| | | LL | 281 | 9.25 | 17.90 | 8.23 | 10.39 | 0.21 | 0.47 | 2.16 | 2.44 | 1.93 | 2.16 | 52.27 | 54.02 |
| | **6-8 steps** | HL | 226 | 8.45 | 12.93 | 6.35 | 6.91 | 0.25 | 0.41 | 2.15 | 2.16 | 1.04 | 1.06 | 15.97 | 16.27 |
| | | LL | 226 | 27.45 | 38.96 | 22.88 | 23.79 | 0.83 | 1.31 | 4.00 | 4.01 | 3.33 | 3.35 | 51.12 | 50.95 |
| | **9-12 steps** | HL | 222 | 10.75 | 13.78 | 6.59 | 6.75 | 0.30 | 0.42 | 2.37 | 2.36 | 1.14 | 1.14 | 14.20 | 14.06 |
| | | LL | 222 | 31.04 | 40.58 | 23.44 | 23.76 | 0.95 | 1.35 | 4.70 | 4.72 | 3.75 | 3.77 | 45.67 | 45.77 |
| | **13+ steps** | HL | 351 | 10.11 | 11.17 | 5.86 | 5.14 | 0.35 | 0.43 | 3.54 | 3.97 | 1.37 | 1.36 | 10.65 | 9.44 |
| | | LL | 351 | 31.10 | 33.47 | 22.52 | 20.93 | 1.08 | 1.28 | 6.05 | 6.25 | 4.57 | 4.74 | 34.03 | 30.75 |
| | **Overall** | H | 1080 | 8.81 | 11.75 | 5.96 | 5.71 | 0.25 | 0.40 | 2.47 | 3.27 | 1.14 | 1.26 | 17.23 | 12.43 |
| | | LL | 1080 | 24.64 | 34.35 | 19.06 | 21.01 | 0.78 | 1.22 | 4.33 | 5.41 | 3.46 | 4.20 | 44.74 | 37.74 |

## F  APPENDIX: ACTION TYPE EXPERIMENTS

Here we provide detailed experimental Results of Agent's performance across different action types in compensation to Section 4.4 Q2 discussions.

Table 6: Model performance on different action types

| Model | Action Type | Eval Level | Count | Type Acc (%) | Exact Acc (%) |
|---|---|---|---|---|---|
| **AGUVIS-7B** | CLICK | LL | 7542 | 99.84 | 91.65 |
| | | HL | 7269 | 67.48 | 34.68 |
| | TYPE | LL | 1122 | 95.28 | 91.89 |
| | | HL | 1063 | 28.69 | 23.52 |
| | SCROLL | LL | 1563 | 99.04 | 87.33 |
| | | HL | 1507 | 14.27 | 11.02 |
| | STOP | LL | 1460 | 86.85 | 86.85 |
| | | HL | 1422 | 0.00 | 0.00 |
| | LONG_PRESS | LL | 74 | 94.59 | 48.65 |
| | | HL | 70 | 1.43 | 1.43 |
| | PRESS | LL | 262 | 93.13 | 93.13 |
| | | HL | 355 | 53.80 | 53.80 |
| | **Overall** | **LL** | **12023** | **97.55** | **90.29** |
| | | **HL** | **11686** | **48.07** | **26.78** |
| **AgentCPM-GUI-7B** | CLICK | LL | 7545 | 97.27 | 88.30 |
| | | HL | 7579 | 86.50 | 54.24 |
| | TYPE | LL | 1105 | 89.32 | 84.98 |
| | | HL | 1121 | 64.67 | 52.45 |
| | SCROLL | LL | 1564 | 98.27 | 96.61 |
| | | HL | 1561 | 45.42 | 41.83 |
| | STOP | LL | 1460 | 79.79 | 79.79 |
| | | HL | 1461 | 66.19 | 66.19 |
| | LONG_PRESS | LL | 73 | 82.19 | 36.99 |
| | | HL | 74 | 2.70 | 1.35 |
| | PRESS | LL | 260 | 29.62 | 29.62 |
| | | HL | 231 | 39.39 | 39.39 |
| | **Overall** | **LL** | **12007** | **92.99** | **86.46** |
| | | **HL** | **12027** | **75.25** | **53.31** |
| **UI-TARS-SFT-7B** | CLICK | LL | 7517 | 99.16 | 79.01 |
| | | HL | 7464 | 88.45 | 54.74 |
| | TYPE | LL | 1124 | 94.40 | 89.15 |
| | | HL | 1121 | 64.32 | 53.35 |
| | SCROLL | LL | 1101 | 99.00 | 94.19 |
| | | HL | 1196 | 52.34 | 47.74 |
| | STOP | LL | 1460 | 99.04 | 99.04 |
| | | HL | 1455 | 44.81 | 44.81 |
| | LONG_PRESS | LL | 74 | 0.00 | 0.00 |
| | | HL | 74 | 0.00 | 0.00 |
| | PRESS | LL | 291 | 53.95 | 53.95 |
| | | HL | 281 | 35.23 | 35.23 |
| | **Overall** | **LL** | **11567** | **96.90** | **82.83** |
| | | **HL** | **11591** | **75.06** | **51.82** |
| **Qwen2.5-VL-7B** | CLICK | LL | 7562 | 93.12 | 83.21 |
| | | HL | 7564 | 67.41 | 44.79 |
| | TYPE | LL | 1124 | 84.34 | 81.49 |
| | | HL | 1123 | 53.25 | 48.00 |

Table 6 – *Continued from previous page*

| Model | Action Type | Eval Level | Count | Type Acc (%) | Exact Acc (%) |
|---|---|---|---|---|---|
| | SCROLL | LL | 1564 | 81.01 | 46.04 |
| | | HL | 1561 | 15.25 | 11.40 |
| | STOP | LL | 1460 | 89.86 | 89.86 |
| | | HL | 1465 | 79.66 | 79.66 |
| | LONG_PRESS | LL | 73 | 60.27 | 50.68 |
| | | HL | 74 | 0.00 | 0.00 |
| | PRESS | LL | 239 | 51.46 | 51.46 |
| | | HL | 231 | 25.54 | 25.54 |
| | **Overall** | **LL** | **12022** | **89.30** | **78.19** |
| | | **HL** | **12018** | **59.59** | **44.36** |
| **OS-Atlas-Pro-7B** | CLICK | LL | 7364 | 94.96 | 74.66 |
| | | HL | 7587 | 86.37 | 48.53 |
| | TYPE | LL | 1054 | 85.67 | 82.16 |
| | | HL | 1123 | 54.50 | 44.43 |
| | SCROLL | LL | 1517 | 48.65 | 28.94 |
| | | HL | 1561 | 29.85 | 26.71 |
| | STOP | LL | 1459 | 67.31 | 67.31 |
| | | HL | 1457 | 35.55 | 35.55 |
| | LONG_PRESS | LL | 71 | 57.75 | 32.39 |
| | | HL | 74 | 0.00 | 0.00 |
| | PRESS | LL | 224 | 72.32 | 72.32 |
| | | HL | 223 | 57.85 | 57.85 |
| | **Overall** | **LL** | **11689** | **84.00** | **68.18** |
| | | **HL** | **12025** | **68.84** | **43.62** |
| **InfiGUI-R1-3B tk** | Click | LL | 7576 | 97.94 | 88.79 |
| | | HL | 7576 | 88.56 | 55.16 |
| | Type | LL | 1125 | 89.78 | 86.84 |
| | | HL | 1125 | 57.24 | 49.33 |
| | Scroll | LL | 1561 | 96.28 | 7.69 |
| | | HL | 1561 | 41.38 | 23.45 |
| | Complete | LL | 1448 | 8.91 | 8.91 |
| | | HL | 1448 | 9.05 | 8.98 |
| | Long Press | LL | 74 | 87.84 | 58.11 |
| | | HL | 74 | 1.35 | 1.35 |
| | Navigate Home | LL | 209 | 90.91 | 0.96 |
| | | HL | 209 | 48.80 | 42.58 |
| | Navigate Back | LL | 22 | 90.91 | 86.36 |
| | | HL | 22 | 54.55 | 18.18 |
| | Wait | LL | 13 | 100.00 | 100.00 |
| | | HL | 13 | 7.69 | 7.69 |
| | Impossible | LL | 1 | 0.00 | 0.00 |
| | | HL | 1 | 0.00 | 0.00 |
| | **Overall** | **LL** | **12029** | **86.04** | **66.76** |
| | | **HL** | **12029** | **68.55** | **44.27** |
| **InfiGUI-R1-3B** | Click | LL | 7576 | 87.04 | 79.14 |
| | | HL | 7576 | 75.17 | 46.57 |
| | Type | LL | 1125 | 85.16 | 82.67 |
| | | HL | 1125 | 54.22 | 47.91 |
| | Scroll | LL | 1561 | 81.81 | 5.64 |
| | | HL | 1561 | 32.42 | 20.24 |
| | Complete | LL | 1448 | 10.43 | 10.43 |
| | | HL | 1448 | 17.47 | 17.40 |

*Continued on next page*

Table 6 – *Continued from previous page*

| Model | Action Type | Eval Level | Count | Type Acc (%) | Exact Acc (%) |
|---|---|---|---|---|---|
| | Long Press | LL | 74 | 79.73 | 47.30 |
| | | HL | 74 | 1.35 | 1.35 |
| | Navigate Home | LL | 209 | 88.52 | 4.31 |
| | | HL | 209 | 45.93 | 39.71 |
| | Navigate Back | LL | 22 | 77.27 | 72.73 |
| | | HL | 22 | 40.91 | 18.18 |
| | Wait | LL | 13 | 100.00 | 100.00 |
| | | HL | 13 | 7.69 | 7.69 |
| | Impossible | LL | 1 | 0.00 | 0.00 |
| | | HL | 1 | 0.00 | 0.00 |
| | **Overall** | **LL** | **12029** | **76.93** | **60.17** |
| | | **HL** | **12029** | **59.61** | **39.27** |

# G  APPENDIX: PLATFORM EXPERIMENTS

Here we provide detailed experimental Results of Agent's performance across different platforms in compensation to Section 4.4 Q2 discussions.

Table 7: Model performance on iOS and Android devices

| Model | Level | Metric | iOS Devices | Android Devices | Overall |
|---|---|---|---|---|---|
| AgentCPM-GUI-8B | LL | EM | 75.49% | 82.96% | 79.32% |
| | | TM | 89.11% | 91.11% | 90.13% |
| | HL | EM | 42.02% | 55.19% | 48.77% |
| | | TM | 68.87% | 80.74% | 74.95% |
| UI-TARS-7B | LL | EM | 73.54% | 86.38% | 79.96% |
| | | TM | 94.16% | 99.22% | 96.69% |
| | HL | EM | 39.30% | 49.02% | 44.14% |
| | | TM | 73.15% | 78.43% | 75.78% |
| Qwen2.5-VL-7B | LL | EM | 66.15% | 71.11% | 68.69% |
| | | TM | 81.32% | 85.93% | 83.68% |
| | HL | EM | 33.85% | 37.04% | 35.48% |
| | | TM | 53.31% | 57.78% | 55.60% |
| OS-Atlas-Pro-7B | LL | EM | 63.89% | 62.31% | 63.08% |
| | | TM | 86.51% | 82.09% | 84.23% |
| | HL | EM | 34.63% | 39.63% | 37.19% |
| | | TM | 64.59% | 66.30% | 65.46% |
| AGUVIS-7B | LL | EM | 82.10% | 88.89% | 85.58% |
| | | TM | 95.72% | 99.26% | 97.53% |
| | HL | EM | 19.43% | 27.17% | 23.44% |
| | | TM | 53.44% | 57.36% | 55.47% |
| InfiGUI-R1-3B thinking | LL | EM | 66.15% | 65.70% | 65.92% |
| | | TM | 81.54% | 90.97% | 86.41% |
| | HL | EM | 32.69% | 37.91% | 35.38% |
| | | TM | 65.77% | 67.87% | 66.85% |
| InfiGUI-R1-3B | LL | EM | 59.23% | 54.87% | 56.98% |
| | | TM | 70.00% | 78.70% | 74.49% |
| | HL | EM | 29.62% | 38.99% | 34.45% |
| | | TM | 56.15% | 64.98% | 60.71% |

## H APPENDIX: LANGUAGE EXPERIMENTS

Here we provide detailed experimental Results of Agent's performance across different languages in compensation to Section 4.4 Q2 discussions.

Table 8: Model performance on Chinese(CN) and English(EN) instructions

| Model Name | Language | Level | TM | EM | EN-CN Difference |
|---|---|---|---|---|---|
| **OS-Atlas-Pro-7B** | CN | HL | 65.09 | 40.88 | TM: +0.36, EM: -0.43 |
| | EN | HL | 65.45 | 40.45 | |
| | CN | LL | 83.68 | 66.86 | TM: 0.00, EM: +2.93 |
| | EN | LL | 83.68 | 69.79 | |
| **Qwen2.5-VL-7B** | CN | HL | 64.53 | 45.76 | TM: -5.96, EM: -2.46 |
| | EN | HL | 58.57 | 43.30 | |
| | CN | LL | 95.11 | 86.14 | TM: -3.09, EM: -4.54 |
| | EN | LL | 92.02 | 81.60 | |
| **AGUVIS-7B** | CN | HL | 47.95 | 26.65 | TM: -4.36, EM: -1.61 |
| | EN | HL | 43.59 | 25.04 | |
| | CN | LL | 96.69 | 88.93 | TM: +0.87, EM: +2.95 |
| | EN | LL | 97.56 | 91.88 | |
| **UI-TARS-7B** | CN | HL | 70.61 | 45.57 | TM: -1.24, EM: +3.01 |
| | EN | HL | 69.37 | 48.58 | |
| | CN | LL | 96.76 | 80.44 | TM: -0.21, EM: +5.67 |
| | EN | LL | 96.55 | 86.11 | |
| **AgentCPM-GUI-8B** | CN | HL | 72.63 | 50.07 | TM: +0.14, EM: +3.02 |
| | EN | HL | 72.77 | 53.09 | |
| | CN | LL | 93.24 | 88.21 | TM: +0.15, EM: +0.29 |
| | EN | LL | 93.39 | 88.50 | |
| **InfiGUI-R1-3B** | CN | HL | 48.64 | 30.96 | TM: +7.39, EM: +6.76 |
| | EN | HL | 56.03 | 37.72 | |
| | CN | LL | 72.84 | 55.17 | TM: +2.16, EM: +3.38 |
| | EN | LL | 75.00 | 58.55 | |
| **InfiGUI-R1-3B thinking** | CN | HL | 65.37 | 41.31 | TM: -2.01, EM: -0.58 |
| | EN | HL | 63.36 | 40.73 | |
| | CN | LL | 87.28 | 65.73 | TM: -3.95, EM: -1.87 |
| | EN | LL | 83.33 | 63.86 | |

