# OpenReview forum: "GAMBIT: A Graph-structured and Decision-Aware Benchmark for MoBile GUI Tasks"
_ICLR.cc/2026/Conference — ICLR 2026 Conference Withdrawn Submission_

### Official Review · Reviewer_baeW · 2025-10-24

**Soundness:** 2
**Presentation:** 3
**Contribution:** 3
**Rating:** 4
**Confidence:** 3

**Summary:**

This paper introduces GAMBIT, a new benchmark designed to evaluate mobile GUI agents on long-horizon, decision-aware tasks across Android and iOS apps. Unlike existing datasets focused on short, linear workflows, GAMBIT includes over 800 task episodes with branching structures and provides dual-level annotations. The authors also propose decision-sensitive evaluation metrics, such as weighted LCS and branch accuracy, and evaluate 7 models, finding sharp performance drops in complex settings. GAMBIT reveals significant challenges in current agents’ ability to reason, plan, and adapt in realistic mobile scenarios.

**Strengths:**

GAMBIT addresses a timely and underexplored challenge—evaluating mobile GUI agents on complex, decision-aware tasks—through the design of a well-constructed benchmark. GAMBIT is notable for its diversity (830 tasks across 35 apps), realistic graph-structured workflows, and dual-level annotations that support both fine-grained and high-level evaluation. The proposed metrics, particularly weighted LCS and decision accuracy, offer a more nuanced view of agent performance beyond traditional step-level success. The experimental evaluation is thorough, spanning seven competitive agents and revealing significant performance gaps that highlight the difficulty and diagnostic value of the benchmark.

**Weaknesses:**

* Although the paper claims that code and data are hosted on an anonymous site, the link was space.

* The experimental results show that all evaluated agents perform poorly on complex decision-aware tasks, with success rates dropping below 5% on longer or branching workflows. While this highlights the benchmark's difficulty, the paper stops short of analyzing *why* agents fail. A more detailed breakdown—e.g., by reasoning failures, perceptual grounding issues, or instruction misinterpretation—would clarify which capabilities current models lack. Additionally, exploring whether training on a subset of the benchmark improves performance on held-out complex tasks would help assess whether these challenges are surmountable with current architectures.

* Although the dataset includes a nontrivial portion of cross-app tasks (~12.5%), the paper does not explicitly analyze how agent performance varies between single-app and cross-app scenarios. Given the increased complexity of app-switching workflows, a focused breakdown would enhance understanding of where current models struggle and how to better design agents for multi-app interactions.

**Questions:**

* Given the extremely low success rates on longer or branching tasks, can the authors provide more detailed analysis of failure cases? Specifically:

   * Are these failures more often due to reasoning errors, perception/grounding problems, or instruction misunderstanding?
   * Could a few concrete examples be shared to illustrate common error modes?
   * Have you considered analyzing errors by task topology (e.g., are hierarchical tasks disproportionately prone to early failure)?

 * Have you considered training or fine-tuning any models on a subset of GAMBIT, such as shorter or less-branching tasks, and evaluating on held-out complex ones? This would help clarify whether current models can improve with exposure, or whether structural architectural innovations are required.

* While GAMBIT includes a nontrivial portion of cross-app tasks, there appears to be no separate evaluation or discussion of how agents perform on these compared to single-app workflows. Could the authors report metrics stratified by cross-app vs. single-app scenarios to better understand the specific difficulties introduced by cross-context transitions?

---

> ### Author Response · Authors · 2025-12-03
> **Response to Reviewer baeW (1/2)**
>
> We greatly value your insightful feedback and acknowledge the necessity for additional details. We appreciate the opportunity to clarify these areas, ensuring our methodology and results are conveyed in the most comprehensive manner.
>
> > ***W1: Although the paper claims that code and data are hosted on an anonymous site, the link was space.***
>
> Thank you for pointing this out! We doubled checked the code and data host on anonymous site and it is available now.
>
> > ***W2: The experimental results show that all evaluated agents perform poorly on complex decision-aware tasks, with success rates dropping below 5% on longer or branching workflows. While this highlights the benchmark's difficulty, the paper stops short of analyzing why agents fail. A more detailed breakdown—e.g., by reasoning failures, perceptual grounding issues, or instruction misinterpretation—would clarify which capabilities current models lack. Additionally, exploring whether training on a subset of the benchmark improves performance on held-out complex tasks would help assess whether these challenges are surmountable with current architectures.***
>
> > ***Q1: Given the extremely low success rates on longer or branching tasks, can the authors provide more detailed analysis of failure cases? Specifically: Are these failures more often due to reasoning errors, perception/grounding problems, or instruction misunderstanding? Could a few concrete examples be shared to illustrate common error modes? Have you considered analyzing errors by task topology (e.g., are hierarchical tasks disproportionately prone to early failure)?***
>
> We are deeply grateful for your valuable suggestions regarding failure cases analysis. We categorize the failure case modes and give concrete examples as follows. From our sampled failure cases, we observe that most errors fall into **two major categories: reasoning-related errors (decision-making weaknesses) and perception-related errors (visual grounding limitations)**. We also noticed that instruction misunderstanding is relatively rare.
>
> For reasoning-related errors:
>
> **Incorrect branch decisions:** We notice that agent sometimes selects the wrong branch when task logic depends on boundary conditions (e.g., choosing the “> 3 items” branch when the desired favorites list contains exactly 3 items).
>
> **Mismatch between reasoning and actions:** the model’s chain-of-thought correctly identifies a condition (e.g., recognizing that a post has >500 likes) but executes the wrong action, such as clicking the “like” button instead of following the intended branch. The model can correctly observe the UI state (e.g., noticing that the notification option is already shown as on/off), but still chooses to click the notification tab to check whether notifications are enabled for no reason.
>
> For perception-related errors:
>
> **Misidentifying UI elements:** The agent may click an incorrect region due to insufficient understanding of on-screen functions (e.g., clicking the article title instead of the “remove from favorites” button).
> **Failure to locate off-screen items:** The agent does not infer that it needs to scroll first, then reveal missing buttons, causing repeated clicks on visible but irrelevant UI components.
> **Task-state misjudgment:** The agent prematurely concludes that a task is complete or fails to recognize completion (e.g., assuming a TV show is already added to My List and skipping the required operation).
>
> When analyzing error by task topology, we find that step-level EM and TM differences across topologies are relatively small. For branch and node decisions, accuracy drops for deeper decision nodes (an average of 5.3% EM drop on later decision nods), indicating that longer dependency chains are more challenging.
>
> We will elaborate these details with more illustrated examples in the revised PDF for a more intuitive failure case study.

---

> ### Author Response · Authors · 2025-12-03
> **Response to Reviewer baeW (2/2)**
>
> > ***Q2: Have you considered training or fine-tuning any models on a subset of GAMBIT, such as shorter or less-branching tasks, and evaluating on held-out complex ones? This would help clarify whether current models can improve with exposure, or whether structural architectural innovations are required.***
>
> Thank you for your suggestions. We conducted a supplementary experiment on GRPO finetuning existing Qwen2.5-VL model with a portion of GAMBIT dataset decision nodes, and evaluate their performance on whole GAMBIT dataset and categorize by decision accuracy. We noticed that GRPO finetuning on these decision-aware actions, increase the first branch decision accuracy by 3.46% and subsequent accuracy by 8.47%. We leave in-depth exploration as subsequent future work.
>
> **Model &nbsp;&nbsp;&nbsp;&nbsp; First Branch &nbsp;&nbsp;&nbsp;&nbsp; Other Branch &nbsp;&nbsp;&nbsp;&nbsp; Total Accuracy**
> ***
>
> Gemini 3 &nbsp; &nbsp; &nbsp; &nbsp; &nbsp; &nbsp; &nbsp; &nbsp; &nbsp; &nbsp; &nbsp; &nbsp; &nbsp; 44.83% &nbsp; &nbsp; 36.24% &nbsp; &nbsp; 40.48%
>
> GPT-5.1 &nbsp; &nbsp; &nbsp; &nbsp; &nbsp; &nbsp; &nbsp; &nbsp; &nbsp; &nbsp; &nbsp; &nbsp; &nbsp; &nbsp; 42.41% &nbsp; &nbsp; 36.39% &nbsp; &nbsp; 39.35%
>
> GPT-4o &nbsp; &nbsp; &nbsp; &nbsp; &nbsp;&nbsp; &nbsp; &nbsp; &nbsp; &nbsp; &nbsp; &nbsp; &nbsp; &nbsp; &nbsp; 28.01% &nbsp; &nbsp; 25.22% &nbsp; &nbsp; 26.59%
>
> Claude-sonnet-4.5 &nbsp; &nbsp; &nbsp; &nbsp; &nbsp;28.53% &nbsp; &nbsp; 23.53% &nbsp; &nbsp; 26.01%
>
> Qwen2.5-VL &nbsp; &nbsp; &nbsp; &nbsp; &nbsp; &nbsp; &nbsp; &nbsp; &nbsp; &nbsp; 37.46% &nbsp; &nbsp; 30.56% &nbsp; &nbsp; 33.95%
>
> **Qwen2.5-VL + GRPO &nbsp; &nbsp; 47.56% &nbsp; &nbsp; 43.06% &nbsp; &nbsp; 45.28%**
>
>
> > ***W3: Although the dataset includes a nontrivial portion of cross-app tasks (~12.5%), the paper does not explicitly analyze how agent performance varies between single-app and cross-app scenarios. Given the increased complexity of app-switching workflows, a focused breakdown would enhance understanding of where current models struggle and how to better design agents for multi-app interactions.***
>
> > ***Q3: While GAMBIT includes a nontrivial portion of cross-app tasks, there appears to be no separate evaluation or discussion of how agents perform on these compared to single-app workflows. Could the authors report metrics stratified by cross-app vs. single-app scenarios to better understand the specific difficulties introduced by cross-context transitions?***
>
> We appreciate your suggestion on evaluating single-app v.s. cross-app performance. We review the performance difference and notice that most models’ GP drop when switching from single-app instructions to cross-app instructions (except for Aguvis). This is due to longer instruction length in cross-app instructions. Current models exhibit little differences in TM and EM, since most instructions are still constructed from basic actions, there are no significant distribution difference in such actions.
>
> We additionally observed a cross-app specific failure mode: when agent switches between two different apps, it must return to the home screen and may need to **swipe to the next page to locate the second app**. Agents may fail to recognize this requirement and repeatedly exploring on the current screen, leading to action termination or navigation failure. We will add these findings in the revised PDF.
>
> We hope that the above response will fully address your concerns. Thank you again for your thorough and helpful review!

---

### Official Review · Reviewer_Esxf · 2025-10-30

**Soundness:** 3
**Presentation:** 3
**Contribution:** 2
**Rating:** 4
**Confidence:** 4

**Summary:**

This paper introduces GAMBIT, a new benchmark for evaluating mobile GUI agents (with more then 800 episodes). Unlike existing benchmarks that primarily focus on simple, linear tasks, GAMBIT presents complex, graph-structured challenges that involve decision-making and branching scenarios to better reflect real-world usage.

**Strengths:**

1. The paper is well-structured and clearly written, making it easy to follow.
2. The contribution of a new, open-source benchmark dataset is a key strength and a valuable resource for the community.
3. The paper provides a thorough experimental evaluation, including comprehensive comparisons with existing agents.

**Weaknesses:**

1. The figures in the paper are of low resolution and are difficult to read.
2. The core innovation appears to be more complex high-level instructions (i.e., adding more conditions and logical judgments), which may not be a substantial enough contribution. The authors need to better justify this.

**Questions:**

1. Fundamental difference from existing datasets: For a "Conditional" or "Hierarchical" episode, are the underlying sequences of screenshots and step-level instructions still inherently sequential? If this is the case, the main distinction of GAMBIT seems to lie only in the complexity of the high-level instruction. Could a similar dataset be constructed by rewriting existing sequential datasets? For instance, by augmenting them with app-specific atomic instructions and constraints, one could transform sequential data into Conditional or Hierarchical tasks. The authors should add a discussion or experiments in the paper to address this point.
2. Unrealistic instructions: In real-world scenarios, users are unlikely to provide such long and detailed instructions. They tend to offer shorter constraints. The example in Figure 1 illustrates this well: a user is more likely to say, "Help me find the cheapest, non-smoking, pet-friendly accommodation in Amsterdam," rather than articulating the complex if-else logic shown in the blue box. The authors need to re-evaluate the plausibility of these high-level instructions. If they do not accurately reflect real user behavior, they should be considered for rewriting to be more naturalistic.
3. About atomic instructions: How many atomic instructions and constraints were generated for each application? This number directly impacts the diversity of the final dataset. If the variety and quantity of these building blocks are insufficient, the dataset's overall diversity will be limited. The authors should provide an analysis and present statistics on this.

---

> ### Author Response · Authors · 2025-12-03
> **Response to Reviewer Esxf (1/2)**
>
> We appreciate the reviewer’s recognition of the clarity of paper presentation, dataset and evaluation. Hereafter, we provide detailed responses to your concerns and questions.
>
> > ***W1: The figures in the paper are of low resolution and are difficult to read.***
>
> We sincerely thank you for pointing this out, we will substitute with high-resolution images and rescale characters larger for reading convenience.
>
>
> > ***W2: The core innovation appears to be more complex high-level instructions (i.e., adding more conditions and logical judgments), which may not be a substantial enough contribution. The authors need to better justify this.***
>
> > ***Q1: Fundamental difference from existing datasets: For a "Conditional" or "Hierarchical" episode, are the underlying sequences of screenshots and step-level instructions still inherently sequential? If this is the case, the main distinction of GAMBIT seems to lie only in the complexity of the high-level instruction. Could a similar dataset be constructed by rewriting existing sequential datasets? For instance, by augmenting them with app-specific atomic instructions and constraints, one could transform sequential data into Conditional or Hierarchical tasks. The authors should add a discussion or experiments in the paper to address this point.***
>
> We appreciate the reviewer’s concern regarding GAMBIT’s fundamental differences from existing datasets, and possibility of extending sequential data to create these complex tasks. We will clarify our motivation and methodological choices as follows.
>
> We acknowledge that a portion of linear instructions could be combined into more complex forms. For example, cross-app instructions could be derived from concatenating two single-app actions (e.g., composing social media apps with entertainment apps with “share to friend” action). Such transformations are not universally feasible for all instructions. Existing mobile GUI datasets typically provide either very fine-grained low-level step instructions or high-level task descriptions without explicit intermediate actions, and **lack well-defined atomic action boundaries** (one atomic action typically include multiple low-level steps). It is challenging to introduce branching conditions or hierarchical structures and maintaining semantic coherency. In contrast, GAMBIT adopts a **workflow-first construction**, decomposing human interactions into atomic actions, and composing them into graph-structured tasks (not merely extending templates or rewriting), thereby modeling complex tasks by graph-structured workflows rather than simply increasing syntactic instruction complexity.
>
> Furthermore, the decision-aware GAMBIT introduce capabilities that cannot be evaluated using purely linear tasks, such as **branch correctness with constraint satisfaction**. These are essential in real-world mobile usage scenarios. We also observed that fine-tuning agents specifically on decision nodes using GRPO, rather than entire instructions, improves decision success rates, grounding ability (e.g., focusing on numerical pricing or rating information on screenshots), and action-type accuracy. This indicates that our introduced workflow structure reveals reasoning behaviors that sequential datasets may not capture. We will revise the paper to clarify these distinctions and better position GAMBIT as a complementary benchmark that targets workflow-level competencies beyond existing datasets.

---

> ### Author Response · Authors · 2025-12-03
> **Response to Reviewer Esxf (2/2)**
>
> > ***Q2: Unrealistic instructions: In real-world scenarios, users are unlikely to provide such long and detailed instructions. They tend to offer shorter constraints. The example in Figure 1 illustrates this well: a user is more likely to say, "Help me find the cheapest, non-smoking, pet-friendly accommodation in Amsterdam," rather than articulating the complex if-else logic shown in the blue box. The authors need to re-evaluate the plausibility of these high-level instructions. If they do not accurately reflect real user behavior, they should be considered for rewriting to be more naturalistic.***
>
> We appreciate your insightful comments on realistic instruction concerns, and we agree that real users typically provide short and vague request. However, **the underlying task workflow to fulfill these requests still remain multi-step (Sequential), conditional decisions (Conjunctive), comparison among alternatives (Selection) and back-and-forth navigation (Hierarchical)**. In practice, even a simple user query (e.g., “Help me find the cheapest, non-smoking, pet-friendly accommodation in Amsterdam”) requires the agent to execute **a sequence of decision-intensive steps** such as applying multiple filters, checking availability, comparing prices, and revisiting pages when results are unsatisfactory. **These behaviors are inherently graph-structured**, even if the user’s direct instruction is short.
>
> The high-level instructions in GAMBIT thus represent an **explicit semantic expansion** of common user intents, making the user requests and actions able to be annotated and evaluated. Additionally, deriving correct branching logic or hierarchical subtasks from a vague instruction is generally unreliable and would not support consistent ground-truth construction. GAMBIT’s instructions are therefore designed to **expose the decision semantics that agents must handle**. Our dataset could also serve as a potential training dataset to compensate for existing non-branching datasets. Our framework can produce shorter paraphrases of these instructions by rewriting, and we will clarify this in the revised PDF to express our intent and design choices more clearly.
>
>
> > ***Q3: About atomic instructions: How many atomic instructions and constraints were generated for each application? This number directly impacts the diversity of the final dataset. If the variety and quantity of these building blocks are insufficient, the dataset's overall diversity will be limited. The authors should provide an analysis and present statistics on this.***
>
> We appreciate your questions regarding detailed statistics. In summary, each instruction in our dataset contains on average 3.51 atomic actions and 4 constraints. Across the full set of 830 instructions, this yields 2,665 atomic actions and 3,037 constraints, reflecting substantial combinational diversity.We will include a detailed breakdown in the revised Appendix to further clarify GAMBIT’s coverage and diversity.

---

### Official Review · Reviewer_Z4tP · 2025-10-30

**Soundness:** 2
**Presentation:** 2
**Contribution:** 2
**Rating:** 4
**Confidence:** 4

**Summary:**

This paper introduces GAMBIT, a new benchmark designed to evaluate mobile GUI agents in complex, graph-structured, and decision-aware task environments. The benchmark features 830 task episodes (totaling over 11,000 actions) across 35 Android and iOS apps, with a variety of task topologies including sequential, conjunctive, conditional, and hierarchical workflows. Tasks are annotated at both high and low levels, and the authors propose new evaluation metrics (weighted LCS and decision accuracy) to more effectively measure long-horizon progress and decision correctness. Extensive experiments on seven current agents demonstrate GAMBIT’s significant difficulty and diagnostic value, exposing key deficiencies in decision-aware and conditional reasoning under mobile GUI interaction.

**Strengths:**

1. GAMBIT advances the field by offering a benchmark that moves decisively beyond template-based and sequential task formulations, emphasizing branching, conditional, and hierarchical dependencies. The graph-based modeling of tasks expands coverage over prior GUI datasets.

2. The benchmark’s 830 graph-structured tasks, spanning multiple app categories, platforms (Android/iOS), and languages (English/Chinese), offer significantly more realistic complexity than previous datasets. Figure 1 and Figure 2 make the contrast between simple sequential templates and graph-structured tasks explicit, showing how real-world constraints and decisions are integrated.

**Weaknesses:**

1.  The methodology and evaluation sections do not discuss or compare against several directly relevant benchmarks from the broader graph learning literature. Notably, there is no mention of GSLB (Li et al., 2023) or GC-Bench (Sun et al., 2024), which set standards for graph-structured benchmarks and could inform both task modeling and metric choices in GAMBIT. Incorporating a principled comparison or at least discussion of these would sharpen the benchmark's positioning and theoretical underpinnings.

2. While the weighted LCS and decision accuracy metrics are described at a high level (Section 3.6, Equation W-LCS), critical formal and implementation details are missing. For example, the exact procedure for weighting within branching workflows (i.e., how weights are propagated on variable-length branches, how conflicting actions in parallel branches are resolved) is not fully specified. For decision accuracy, the criteria for identifying branch points and their correct traversal could be formalized, perhaps as a function $D(\hat{\mathcal{T}}, \mathcal{G})$ given the prediction $\hat{\mathcal{T}}$ and gold graph $\mathcal{G}$. This lack of mathematical rigor makes reproducibility and external comparison more difficult and reduces trust in the fairness of new metrics.

3. While the empirical pipeline is thorough, the paper does not attempt to theoretically ground the choice of task structures, guard conditions, or evaluation metrics. For instance, it is unclear whether the four chosen graph topologies (Figure 2(b)) cover the space of real-world app workflows (or could be unified under a broader formalism). There is also no formal complexity analysis (e.g., task branching factor distribution), nor a justification (proof or constructivist argument) of metric sensitivity or discriminative power over prior best practices (EM, SR, GP).

4. Insufficient comparison/enumeration of alternative metrics: Although GAMBIT introduces new metrics, the paper under-delivers on a rationale for why the proposed design is preferable to other sequence/graph alignment metrics (e.g., edit distance, graph isomorphism, tree edit distance), or whether decision accuracy is robust to minor label annotation inconsistencies. This leaves open whether the evaluation suite truly advances the measurement of agent competence.

5. Table 4 and appendix tables reveal stark differences in per-action model performance (e.g., on Long Press and Complete/Stop). Yet there's little explanation or modeling of possible action ambiguity (multiple equivalent ways to complete a task), nor an attempt to quantify inter-annotator agreement on low-level/stepwise execution.

6. For tasks involving decisions or fallback paths (hierarchical/conditional), the sampling procedure for branches (e.g., negative/”IMPOSSIBLE” cases) is only cursorily discussed. How are ambiguous, unreachable, or failed state paths annotated and incorporated in analysis? Are rejected paths equally weighted when scoring W-LCS or decision accuracy?

**Questions:**

N/A

---

> ### Author Response · Authors · 2025-12-03
> **Response to Reviewer Z4tP (1/2)**
>
> > ***W1: The methodology and evaluation sections do not discuss or compare against several directly relevant benchmarks from the broader graph learning literature. Notably, there is no mention of GSLB (Li et al., 2023) or GC-Bench (Sun et al., 2024), which set standards for graph-structured benchmarks and could inform both task modeling and metric choices in GAMBIT. Incorporating a principled comparison or at least discussion of these would sharpen the benchmark's positioning and theoretical underpinnings.***
>
> We appreciate the reviewer’s suggestion on mentioning GSLB, GC-Bench and potential benchmarks evaluating graph representation models. However, we respectfully note that the “graph structure” in GAMBIT serve a fundamentally different purpose: **these graph topologies are used to reconstruct mobile GUI agent workflows from atomic actions.** These workflow graphs are not data graphs used for training or evaluating Graph Neural Networks (GNNs), and they do not involve node or edge features, graph embeddings or graph-level prediction tasks. The goal of GNN evaluation datasets (graph classification, link prediction, etc.) differ substantially from our chosen GUI agent benchmarks (GUI element perception and grounding, agent task decision, etc.) and these targets are orthogonal. We would include a brief clarification in the revised Related Work section to avoid confusion.
>
>
> > ***W2: While the weighted LCS and decision accuracy metrics are described at a high level (Section 3.6, Equation W-LCS), critical formal and implementation details are missing. For example, the exact procedure for weighting within branching workflows (i.e., how weights are propagated on variable-length branches, how conflicting actions in parallel branches are resolved) is not fully specified. For decision accuracy, the criteria for identifying branch points and their correct traversal could be formalized, perhaps as a function given the prediction and gold graph . This lack of mathematical rigor makes reproducibility and external comparison more difficult and reduces trust in the fairness of new metrics.***
>
> We appreciate your suggestion to clarify the metric definitions, in our mobile GUI agent setting, the annotated gold label and agent outputs are action sequences, even when their underlying tasks workflow contain branching logic. Therefore, W-LCS does not involve weighting propagation or conflict resolution over graph structures. Following prior works’ evaluation settings, for decision accuracy, branch points are explicitly annotated in GAMBIT’s tasks and are evaluated as simple categorical decisions (for a single-step low-level action, 0 denotes failure and 1 denotes success). We will revise the paper to present these definitions more clearly within the GUI-agent evaluation context.
>
>
> > ***W3: While the empirical pipeline is thorough, the paper does not attempt to theoretically ground the choice of task structures, guard conditions, or evaluation metrics. For instance, it is unclear whether the four chosen graph topologies (Figure 2(b)) cover the space of real-world app workflows (or could be unified under a broader formalism). There is also no formal complexity analysis (e.g., task branching factor distribution), nor a justification (proof or constructivist argument) of metric sensitivity or discriminative power over prior best practices (EM, SR, GP).***
> > ***W4: Insufficient comparison/enumeration of alternative metrics: Although GAMBIT introduces new metrics, the paper under-delivers on a rationale for why the proposed design is preferable to other sequence/graph alignment metrics (e.g., edit distance, graph isomorphism, tree edit distance), or whether decision accuracy is robust to minor label annotation inconsistencies. This leaves open whether the evaluation suite truly advances the measurement of agent competence.***
>
> We appreciate your thoughtful comments. In our mobile GUI agent settings, the task structures and guard conditions are typically empirically grounded from everyday mobile app user patterns and interactions, rather than from theoretical notions of graph completeness in a GNN view. The four workflows we adopt reflect the decision-making procedures seen across real-world apps, and are intended as a compensation for existing mobile GUI benchmarks, not an exhaustive taxonomy of all possible app workflows. In addition to the mobile GUI agent commonly adopted metrics (TM, EM, SR, GP), W-LCS (an extension of previous metrics) is designed to provide additional sensitivity to long-horizon task workflows and branching decisions. The effectiveness is demonstrated empirically through model comparisons. We will elaborate these details and design principles in the GUI-agent evaluation context in revised PDF.

---

> ### Author Response · Authors · 2025-12-03
> **Response to Reviewer (2/2)**
>
> > ***W5: Table 4 and appendix tables reveal stark differences in per-action model performance (e.g., on Long Press and Complete/Stop). Yet there's little explanation or modeling of possible action ambiguity (multiple equivalent ways to complete a task), nor an attempt to quantify inter-annotator agreement on low-level/stepwise execution.***
>
> We appreciate your concerns regarding action ambiguity and inter-annotator agreement. Action ambiguity is a common issue across existing mobile GUI benchmarks (e.g., AITW, AITZ, GUI-Odyssey), to mitigate this:
>
> (1) We removed tasks exhibiting substantial ambiguity during annotation and quality control.
>
> (2) We asked annotators to adopt the most natural and human-preferred action sequence when only minor variants existed.
>
> (3) Our decision ware W-LCS metric tolerates minor step-order variant and reduce false negatives during evaluation (compared to SR or GP, W-LCS calculates the common sequence length between agent prediction and golden label).
>
> Regarding inter-annotator agreement, (1) all annotators underwent unified training, (2) we provided a standardized library of low-level action templates, (3) two senior annotators performed cross-checks on all trajectories and resolved substantial disagreements. We will clarify these procedures in the revised PDF.
>
>
> > ***W6: For tasks involving decisions or fallback paths (hierarchical/conditional), the sampling procedure for branches (e.g., negative/”IMPOSSIBLE” cases) is only cursorily discussed. How are ambiguous, unreachable, or failed state paths annotated and incorporated in analysis? Are rejected paths equally weighted when scoring W-LCS or decision accuracy?***
>
> Similar to prior mobile GUI datasets, our trajectories are constructed from successful gold executions, unreachable or failed paths are relatively rare in our practice. In our case, “Impossible” arise when all task branches with conditions are unsatisfiable. During annotation and evaluation, **we treat “Impossible” as a regular action type** and it is weighted identically to other low-level actions (e.g. Press, Scroll, etc.). We will include a complete action-type distribution in the revised PDF.

---

### Official Review · Reviewer_Tuv1 · 2025-11-01

**Soundness:** 3
**Presentation:** 2
**Contribution:** 2
**Rating:** 4
**Confidence:** 4

**Summary:**

This paper proposes GAMBIT, a graph-structured and decision-aware benchmark designed for evaluating mobile GUI agents on long-horizon and complex tasks. It features diverse graph topologies, covers both Android and iOS platforms, and includes cross-application scenarios. It also proposes novel evaluation metrics beyond task success rate and step accuracy.

**Strengths:**

- The focus on long-horizon, decision-aware complex tasks is well-motivated and addresses a clear gap in existing mobile GUI benchmarks.

- Representing complex tasks using graph structures is innovative.

- The benchmark is comprehensive, covering cross-platform (Android and iOS) and cross-app scenarios.

**Weaknesses:**

- The benchmark remains offline and static. It is unclear how it handles evaluation scenarios where multiple valid action trajectories or paths exist for completing a task, which is common in real-world GUI interactions.

- The experimental evaluation lacks results from state-of-the-art closed-source models (e.g., Claude, Gemini), limiting the analysis of actual task difficulty

- Many descriptions in the paper are unclear and ambiguous. For instance, the "dual-layer quality control" is mentioned in the paper but not elaborated on subsequently.

**Questions:**

- How does the benchmark's graph architecture account for tasks where multiple valid trajectories can lead to successful completion? Are all valid paths considered in the ground truth or metrics?

- Could you please clarify what the "dual-layer" in "dual-layer quality control" means?

- The benchmark is claimed to be representative of everyday usage but lacks justification, especially compared with other other benchmarks like AndroidWorld[1], SPA-bench[2].

[1] Rawles C, Clinckemaillie S, Chang Y, et al. Androidworld: A dynamic benchmarking environment for autonomous agents[J]. arXiv preprint arXiv:2405.14573, 2024.

[2] Chen J, Yuen D, Xie B, et al. Spa-bench: A comprehensive benchmark for smartphone agent evaluation[C]//NeurIPS 2024 Workshop on Open-World Agents. 2024.

---

> ### Author Response · Authors · 2025-12-03
> **Response to Reviewer Tuv1 (1/2)**
>
> We sincerely thank the reviewer for the positive feedback on our task motivation, graph structure innovation and benchmark coverage. We appreciate the opportunity to clarify specific concerns.
>
> > ***W1: The benchmark remains offline and static. It is unclear how it handles evaluation scenarios where multiple valid action trajectories or paths exist for completing a task, which is common in real-world GUI interactions.***
>
> > ***Q1: How does the benchmark's graph architecture account for tasks where multiple valid trajectories can lead to successful completion? Are all valid paths considered in the ground truth or metrics?***
>
> We appreciate the reviewer’s concern. Offline and static evaluation currently remains standard practice in existing mobile GUI benchmarks, as it provides **reproducible, consistent, and controlled conditions** for quantitative comparison across agents and datasets. We also acknowledge that mobile GUI may allow multiple valid action trajectories. This challenge is also common across prior benchmarks. To mitigate this issue:
>
> (1) We removed tasks with substantial ambiguity during manual annotation and quality checking.
>
> (2) For minor variants, annotators selected the most natural and human-preferred actions.
>
> (3) Our W-LCS metric helps reduce potential false negatives by matching subsequences rather than requiring exact step-by-step alignment (e.g., GP or SR).
>
>
> > ***W2: The experimental evaluation lacks results from state-of-the-art closed-source models (e.g., Claude, Gemini), limiting the analysis of actual task difficulty.***
>
> Thank you for your suggestions, we conducted supplementary experimental evaluations with closed-source models including: Claude, Gemini-3, GPT-5, GPT-4o. We report their performance with Qwen2.5-VL (already reported in paper Table 3) as a comparison. We will elaborate these details in the revised PDF
>
> **Model (Level) &nbsp; &nbsp; &nbsp; &nbsp; &nbsp; &nbsp; &nbsp; &nbsp; &nbsp;Single (EM/TM/SR/GP) &nbsp; &nbsp; &nbsp; &nbsp;  &nbsp; &nbsp; Conj (EM/TM/SR/GP) &nbsp; &nbsp; &nbsp; &nbsp; &nbsp; &nbsp; &nbsp; &nbsp; Seq (EM/TM/SR/GP) &nbsp; &nbsp; &nbsp; &nbsp; &nbsp; &nbsp; Cond (EM/TM/SR/GP) &nbsp; &nbsp; &nbsp; &nbsp; Hier (EM/TM/SR/GP)**
> ***
>
> Qwen2.5-VL-7B (LL) ｜ 79.90 / 91.21 / 62.80 / 59.99｜81.26 / 92.98 / 36.14 / 51.76｜76.62 / 87.94 / 24.19 / 46.25｜80.44 / 91.65 / 19.10 / 51.61｜75.68 / 86.08 / 11.18 / 48.32
>
> Qwen2.5-VL-7B (HL)｜58.55 / 73.53 / 38.00 / 35.66｜53.66 / 71.43 / 5.45 / 17.45｜44.80 / 59.01 / 4.33 / 18.70｜41.82 / 57.53 / 1.01 / 19.54｜36.31 / 50.99 / 0.66 / 16.02｜
>
> GPT-5.1 (LL) &nbsp; &nbsp; &nbsp; &nbsp; &nbsp; &nbsp; &nbsp;｜48.60 / 93.80 / 11.60 / 20.25｜39.41 / 93.45 / 1.98 / 12.24｜42.24 / 95.17 / 2.17 / 10.14｜46.70 / 94.29 / 0.50 / 13.35｜44.28 / 94.26 / 0.00 / 9.92
>
> GPT-5.1 (HL) &nbsp; &nbsp; &nbsp; &nbsp; &nbsp; &nbsp; &nbsp;｜38.43 / 80.54 / 6.00 / 12.76｜32.00 / 80.81 / 0.00 / 6.16｜32.86 / 80.16 / 0.00 / 4.52｜34.24 / 72.66 / 0.00 / 8.36｜33.07 / 75.11 / 0.00 / 4.73
>
> GPT-4o (LL) &nbsp; &nbsp; &nbsp; &nbsp; &nbsp; &nbsp; &nbsp; &nbsp;｜36.65 / 86.14 / 6.00 / 11.23｜30.83 / 88.35 / 0.99 / 5.12｜29.64 / 85.26 / 0.00 / 4.48｜30.14 / 81.56 / 0.00 / 4.93｜33.06 / 82.46 / 0.00 / 6.04
>
> GPT-4o (HL)&nbsp; &nbsp; &nbsp; &nbsp; &nbsp; &nbsp; &nbsp; &nbsp;｜29.23 / 68.68 / 3.20 / 7.35｜25.15 / 76.27 / 0.00 / 3.42｜26.26 / 75.13 / 1.08 / 3.31｜25.11 / 66.94 / 0.00 / 3.79｜24.45 / 68.95 / 0.00 / 3.30
>
> Claude-sonnet-4.5 (LL)｜37.43 / 88.84 / 10.04 / 5.58｜27.66 / 89.72 / 0.99 / 4.97｜27.57 / 91.61 / 1.44 / 3.53｜31.65 / 91.63 / 0.50 / 4.44｜30.19 / 91.39 / 0.00 / 3.18
>
> Claude-sonnet-4.5 (HL)｜27.08 / 77.96 / 5.67 / 3.71｜21.93 / 81.17 / 0.99 / 3.27｜23.36 / 90.96 / 0.36 / 2.33｜24.13 / 75.54 / 0.00 / 4.39｜22.37 / 76.60 / 0.00 / 2.60
>
> Gemini 3 Pro Preview (LL)｜57.13 / 90.82 / 18.00 / 28.36｜52.04 / 93.95 / 3.96 / 15.92｜56.94 / 95.09 / 0.72 / 12.18｜58.90 / 93.94 / 0.50 / 17.49｜56.63 / 93.50 / 0.00 / 13.02
>
> Gemini 3 Pro Preview (HL)｜51.86 / 82.71 / 18.40 / 27.11｜40.62 / 83.45 / 0.99 / 9.37｜43.67 / 81.95 / 0.36 / 7.54｜42.39 / 78.66 / 0.00 / 12.18｜39.13 / 79.30 / 0.00 / 7.34
>
>
>
> > ***W3: Many descriptions in the paper are unclear and ambiguous. For instance, the "dual-layer quality control" is mentioned in the paper but not elaborated on subsequently.***
>
> > ***Q2: Could you please clarify what the "dual-layer" in "dual-layer quality control" means?***
>
> Thank you for pointing this out. The “dual-layer quality control” refers to two human-verification steps in our dataset construction pipeline: (1) during graph structured task modeling (Fig. 2 (b)), where human reviewers validate the correctness and executability of atomic actions, graph structured tasks and complex high-level instructions; and (2) during annotation (Fig. 2 (d)), where annotators cross-check trajectory and instructions to ensure consistency and overall quality. We agree that this description should be made clearer, and we will revise the PDF to explain more explicitly, and maintain consistent wording throughout the paper.

---

> ### Author Response · Authors · 2025-12-03
> **Response to Reviewer Tvu1 (2/2)**
>
> > ***Q3: The benchmark is claimed to be representative of everyday usage but lacks justification, especially compared with other benchmarks like AndroidWorld, SPA-bench.***
>
> We appreciate the reviewer’s question regarding everyday mobile usage. Our goal is to **bring GAMBIT closer to everyday usage patterns**, particularly those involving complex, long-horizon and decision-aware workflows (e.g., filtering products by price constraints, making hotel choices by rating comparison and weather conditions, etc.). Along this dimension, GAMBIT **compensates for these aspects not emphasized** in prior benchmarks, which focus on broader mobile app coverage but shorter, linear tasks.
>
> To model these usage patterns, we construct tasks by decomposing human mobile interactions into **atomic actions** and composing them into **graph-structured workflows** to support hierarchical subtasks and conditional branching. Compared with SPA-Bench and GUI-Odyssey, which expand instructions through template-based substitution or incremental clause addition, our approach focuses on how users perform multi-step decision-making in real apps. We will revise the manuscript to clarify our intended meaning to situate GAMBIT more clearly as **complementary** to existing benchmarks.
>
> We sincerely appreciate your invaluable feedback and we are dedicated to refining this work based on your respected suggestions.

---

### Note · Authors · 2026-01-24

I have read and agree with the venue's withdrawal policy on behalf of myself and my co-authors.